



# Active tectonic field for CO$_2$ Storage management: Hontomín onshore study-case (SPAIN)

Raúl Pérez-López[1], José F. Mediato[1], Miguel A. Rodríguez-Pascua[1], Jorge L. Giner-Robles[2], Adrià Ramos[1], Silvia Martín-Velázquez[3], Roberto Martínez-Orío[1], Paula Fernández-Canteli[1]

1. IGME – Instituto Geológico y Minero de España – Geological Survey of Spain. C/Ríos Rosas 23, Madrid 28003 – SPAIN. Email: r.perez@igme.es, jf.mediato@igme.es, ma.rodriguez@igme.es, ro.martinez@igme.es, a.ramos@igme.es; paula.canteli@igme.es
2. Departamento de Geología y Geoquímica. Facultad de Ciencias. Universidad Autónoma de Madrid. Campus Cantoblanco, Madrid. SPAIN. Email: jorge.giner@uam.es
3. Universidad Rey Juan Carlos. Email: silvia.martin@urjc.es

*Abstract*

*One of the concerns of underground CO$_2$ onshore storage is the triggering of Induced Seismicity and fault reactivation. Hence, a comprehensive analysis of the tectonic parameters involved in the storage rock formation is mandatory for safety management operations. Unquestionably, active faults and seal faults depicting the storage bulk are relevant parameters to be considered. However, there is a lack of analysis of the active tectonic strain field affecting these faults during the CO$_2$ storage monitoring. The advantage of reconstructing the tectonic field is the possibility to determine the strain trajectories and describing the fault patterns affecting the reservoir rock. In this work, we adapt a methodology of systematic geostructural analysis to the underground CO$_2$ storage, based on the calculation of the strain field and defined by the strain field from kinematics indicators on the fault planes (e$_y$ and e$_x$ for the maximum and minimum horizontal shortening respectively),. This methodology is based on a statistical analysis of individual strain tensor solutions obtained from fresh outcrops. Consequently, we have collected 447 fault data in 32 field stations located within a 20 km radius. The understanding of the fault sets role for underground fluid circulation can also be established, helping for further analysis of CO$_2$ leakage and seepage. We have applied this methodology to Hontomín onshore CO$_2$ storage facilities (Central Spain). The*



*geology of the area and the number of high-quality outcrops made this site as a good*
*candidate for studying the strain field from kinematics fault analysis. The results*
*indicate a strike-slip tectonic regime with the maximum horizontal shortening with*
*N160°E and N50ºE trend for the local regime, which activates NE-SW strike-slip faults*
*and NE-SW compressional faults. A regional tectonic field was also recognized with a*
*N-S trend, which activates E-W compressional faults. Monitoring of E-W faults within*
*the reservoir is suggested in addition to the possibility of obtaining focal mechanism*
*solutions for microearthquakes (M < 3).*
Keywords: onshore $CO_2$ storage, tectonic field, paleostrain analysis, active fault,
Hontomín onshore pilot-plant.
1. INTRODUCTION
Industrial made-man activities generate $CO_2$ that could change the chemical balance of
the atmosphere and their relationship with the geosphere. Carbon capture and
sequestration (CCS) appears as a good choice to reduce the $CO_2$ gas emission to the
atmosphere (Christensen, 2004), allowing the industry increasing activity with a low
pollution impact. There is a lot of literature about what must have a site to be a potential
underground storage suitable to CCS (e.g. Chu, 2009; Orr, 2009; Goldberg et al., 2010
among others). The reservoir sealing, the caprock, permeability and porosity, plus
injection pressure and volume injected, are the main considerations to choose one
geological subsurface formation as the $CO_2$ host-rock. In this frame, the tectonic active
field is considered in two principal ways: (1) to prevent the fault activation and
earthquakes triggering, with the consequence of leakage and seepage, and (2) the long-
term reservoir behavior, understanding as long-term from centennial to millennial time-



span. Therefore, what is the long-term behavior of CCS? What do we need to monitor
for a safe CCS management? Winthaegen et al. (2005) suggest three subjects for
monitoring: (a) the atmosphere air quality near the injection facilities, due to the $CO_2$
toxicity (values greater than 4%, see Rice, 2003 and Permentier et al., 2017), (b) the
overburden monitoring faults and wells and (c) the sealing of the reservoir. The study of
natural analogues for CCS is a good strategy to estimate the long-term behavior of the
reservoir, considering parameters as the injected $CO_2$ pressure and volume, plus the
brine mixing with $CO_2$ (Pearce, 2006). Hence, the prediction of site performance over
long timescales also requires an understanding of $CO_2$ behavior within the reservoir, the
mechanisms of migration out of the reservoir, and the potential impacts of a leak on the
near surface environment. The assessments of such risks will rely on a combination of
predictive models of $CO_2$ behavior, including the fluid migration and the long-term
$CO_2$-porewater-mineralogical interactions (Pearce, 2006). Once again, the tectonic
active field interacts directly on this assessment. Moreover, the fault reactivation due to
the pore pressure increasing during the injection and storage has also to be considered
(Röhmann et al., 2013). Despite the uplift measure by Röhmann et al. (2013) are
submillimeter (c.a. 0.021 mm) at the end of the injection processes, given the ongoing
occurrence of microearthquakes, long-term monitoring is required. The geomechanical
and geological models predict the reservoir behavior and the caprock sealing properties.
The role of the faults inside these models is crucial for the tectonic long-term behavior
and the reactivation of faults that could trigger earthquakes.
Concerning the Induced Seismicity, Wilson et al. (2017) published the Hi-Quake
database, with a classification of all man-made earthquakes according to the literature,
in an online repository (https://inducedearthquakes.org/, last access on May, 2019). This
database includes 834 projects with proved Induced Seismicity, where two different





cases with earthquakes as larger as M 1.7, detected in swarms about 9,500
microearthquakes, are related to CCS operations. Additionally, Foulger et al. (2018)
pointed out that CCS can trigger earthquakes with magnitudes lesser than M 2, namely
the cases described in their work are as greater as M 1.8, with the epicenter location 2
km around the facilities. McNamara (2016) described a comprehensive method and
protocol for monitoring CCS reservoir for the assessment and management of Induced
Seismicity. The knowledge of active fault patterns and the stress/strain field could help
on designing monitoring network and identifying those faults capable for triggering
micro-earthquakes (M < 2) and/or breaking the sealing for leakage (patterns of open
faults for low-permeability $CO_2$ migration).
In this work, we propose that the description, the analysis and establishment of the
tectonic strain field have to be mandatory for long-term CCS monitoring and
management, implementing the fault behavior in the geomechanical models. This
analysis does not increase the cost for long-term monitoring, given that they are low-
cost and the results are acquired in a few months. Therefore, we propose a methodology
based on the reconstruction of the strain field from the classical studies in geodynamics
(Angelier, 1979 and 1984; Reches, 1983; Reches, 1987). As a novelty, we introduce the
strain fields (SF) between 20 away from the subsurface reservoir deep geometry. The
knowledge of the strain field at local scale allows to classify the type of faulting and
their role for leakage processes, whilst the regional scale explores the tectonic active
faults which could affect the reservoir. The methodology is rather simple, taking
measures of slickensides and striations on fault planes to establish the orientation of the
maximum horizontal shortening ($e_y$), and the minimum horizontal shortening ($e_x$) for
the strain tensor. The principal advantage of the SF analysis is the directly classification





of all the faults involved into the geomechanical model and the prediction of the failure
parameters.
The tectonic characterization of the CCS of Hontomín was implemented in the
geological model described by Le Gallo and de Dios (2018). Beyond the use of Induced
Seismicity and potentially active faults, the scope of this method is to propose an initial
protocol to manage underground storage operations.

2. HONTOMÍN ONSHORE STUDY CASE
*2.1 Geological description of the reservoir*
The $CO_2$ storage site of Hontomín is enclosed in the southern section of the Mesozoic
Basque–Cantabrian Basin, known as Burgalesa Platform (Serrano and Martinez del
Olmo, 1990, Tavani, 2012), within the sedimentary Bureba Basin (**Fig. 1**). This
geological domain is located in the northern junction of the Cenozoic Duero and Ebro
basins, forming an ESE-dipping monocline bounded by the Cantabrian Mountains
Thrust to the north, the Ubierna Fault System (UFS) to the south and the Asturian
Massif to the west (**Fig. 1**).
The Meso-Cenozoic tectonic evolution of the Burgalesa Platform starts with a first rift
period during Permian and Triassic times (Calvet et al., 2004; Dallmeyer and Martínez-
García, 1990), followed by a relative tectonic quiescence during Early and Middle
Jurassic times (e.g. Aurell et al., 2002). The main rifting phase took place during the
Late Jurassic and Early Cretaceous times, due to the opening of the North Atlantic and
the Bay of Biscay-Pyrenean rift system (García-Mondéjar et al., 1986; García-
Mondéjar et al., 1996; Le Pichon and Sibuet, 1971; Lepvrier and Martínez-García,
1990; Roca et al., 2011; Tugend et al., 2014). The convergence between Iberia and
Eurasia from Late Cretaceous to Miocene times triggered the inversion of previous

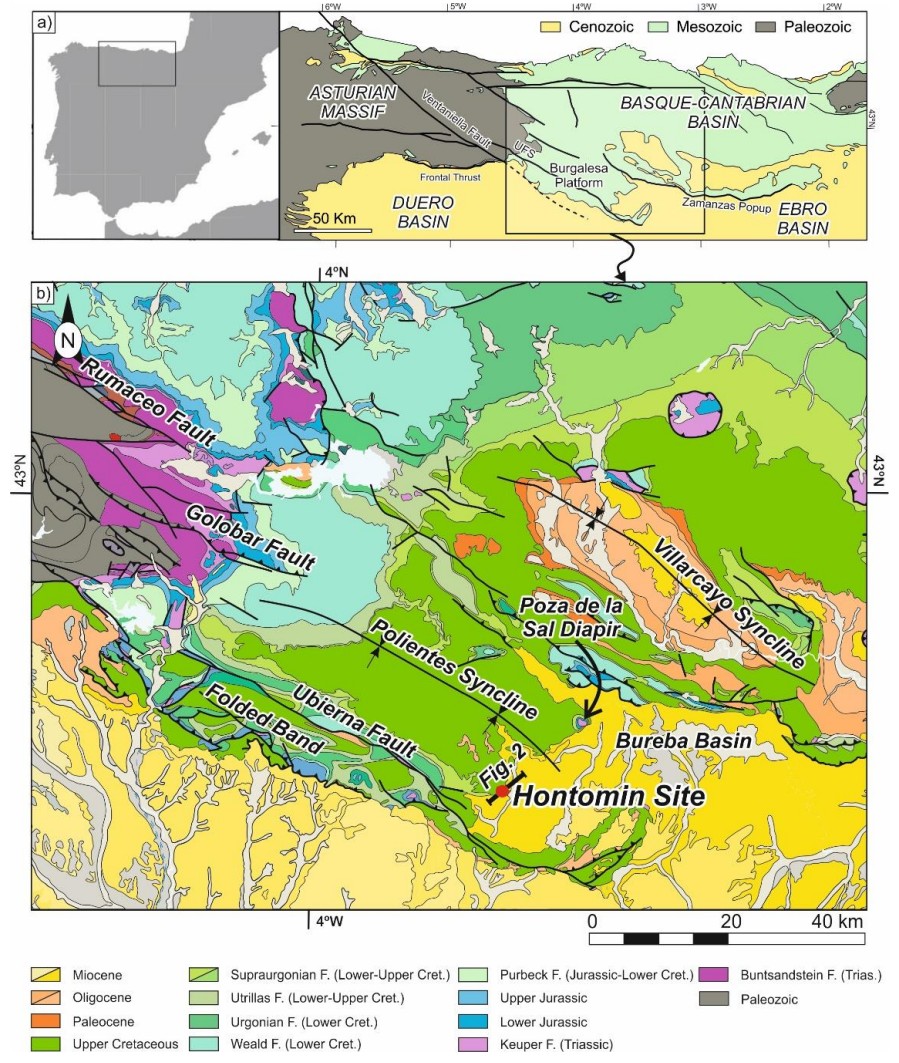



*Figure 1. a) Location map of the study area in the Iberian Peninsula, along with the geological map of*

*the Asturian and Basque-Cantabrian areas, labelling major units and faults (modified after Quintà and*

*Tavani 2012); b) Geographical location of Hontomín pilot-plant (red dot) within the Basque-Cantabrian*

*Basin. This basin is tectonically controlled by the Ubierna Fault System (UFS; NW-SE oriented) and the*

*parallel Polientes syncline, the Duero and Ebro Tertiary basins and Poza de la Sal evaporitic diapir.*

*Cret: Cretaceous; F: Facies.*


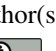




Mesozoic extensional faults and the development of a E-W orogenic belt (Cantabrian
domain to the west and Pyrenean domain to the east) formed along the northern Iberian
plate margin (Gómez et al., 2002; Muñoz, 1992; Vergés et al., 2002).
The facilities are located within the Basque-Cantabrian Basin (**Fig. 1b**). This reservoir
is a deep saline aquifer developed in fractured Jurassic carbonates, with a low porous
permeability matrix, located at almost 1,500 m depth (Alcalde et al., 2013). The
Hontomín geological structure was described by Alcalde et al. (2013) and Le Gallo and
de Dios (2018), and defined as a fold-related dome (Tavani et al., 2013). The reservoir
structure is associated to the Cenozoic extensional tectonic stages, according to these
authors. The present-day geological structure is related with the reactivation by a
tectonic compressional phase during the Cenozoic Pyrenees compression (Alcalde et al.,

152    2013).

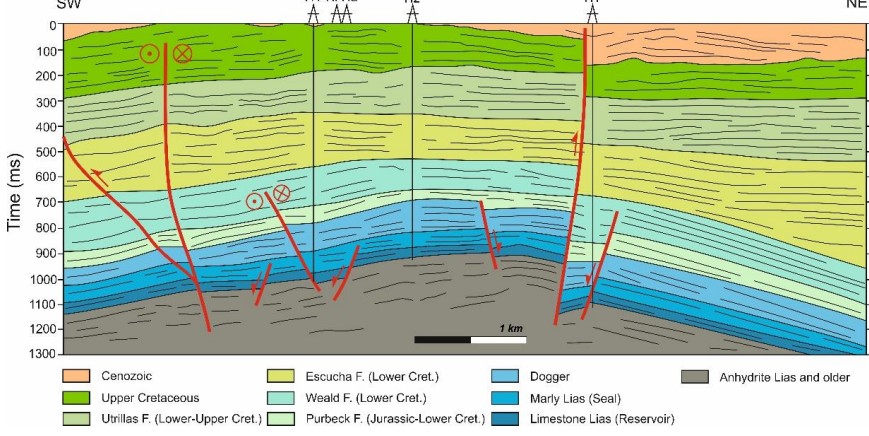



*Figure 2: Interpretation of a 2D seismic reflection profile crossing the oil exploration wells (H1, H2 and*
*H4), along with the monitoring well (Ha) and injection well (Hi) through Hontomin Pilot Plant (HPP).*
*Modified from Alcalde et al. (2014). See Figure 1 for location, black line at the red circle.*





The Hontomín structure is bordered by the UFS to the south and west, by the Poza de la
Sal diapir and the Zamanzas Popup structure (Carola, 2014) to the north and by the
Ebro Basin to the east (**Fig. 1**). The structure has been classified as forced fold related
dome structure (Tavani et al., 2013; **Fig. 2**), which was formed by an extensional fault
system with migration of evaporites towards the hanging wall during the Mesozoic
(Soto et al., 2011). During the tectonic compressional phase, associated with Cenozoic
tectonics affecting the Pyrenees, the right-lateral transpressive inversion of the basement
faults was activated, plus the reactivation of transverse extensional faults (**Fig. 2**; Tavani
et al., 2013; Alcalde et al., 2014).

At the HPP, the target reservoir and seal formations consist of Lower Jurassic marine
carbonates, arranged in an asymmetric dome-like structure (**Fig. 2**) with an overall
extent of 15 km$^2$ and located at 1,485 m of depth (Alcalde et al., 2013, 2014; Ogaya et
al., 2013). The target $CO_2$ injection point is a saline aquifer formed by a dolostone unit,
known as "Carniolas", and an oolitic limestone of the Sopeña Formation, both
corresponding to Lias in time (Early Jurassic). The estimated porosity of the Carniolas
reaches over 12% (Ogaya et al., 2013; Le Gallo and de Dios, 2018) and it is slightly
lower at the Carbonate Lias level (8.5% in average). The reservoir levels contain saline
water with more than 20 g/l of NaCl and very low oil content. The high porosity of the
lower part of the reservoir (i.e., the Carniolas level) is the result of secondary
dolomitization and different fracturing events (Alcalde et al., 2014). The minimum
thickness of the reservoir units is 100 m. The potential upper seal unit comprises Lias
marlstones and black shales from a hemipelagic ramp (**Fig. 2**); Pliensbachian and
Toarcian) of the "Puerto del Pozazal" and Sopeña Formations.





*2.2 Regional tectonic field*
The tectonic context of HPP has been described from two different approaches: (1) the
tectonic style of the fractures bordering the Hontomín reservoir (De Vicente et al., 2011;
Tavani et al., 2011) and (2) the tectonic regional field described from earthquakes with
mechanism solutions and GPS data (Herraiz et al., 2000; Stich et al., 2006; De Vicente
et al., 2008).
(1) The tectonic style of the Bureba Basin was described by De Vicente et al. (2011),
which classified the Basque-Cantabrian Cenozoic Basin (**Fig. 1a**) as transpressional
with contractional horsetail splay basin. The NW-SE oriented Ventaniella fault (**Fig.**
**1a**), includes the UFS in the southeastward area, being active between the Permian and
Triassic period, and strike-slip during the Cenozoic contraction. In this tectonic
configuration, the Ubierna Fault is a right-lateral strike-slip fault. These authors pointed
out the sharp contacts between the thrusts and the strike-slip faults in this basin.
Furthermore, Tavani et al. (2011) also described complex Cenozoic tectonic context
where right-lateral tectonic style reactivated WNW-ESE trending faults. Both the
Ventaniella and the Ubierna faults acted as transpressive structures forming 120 km
long and 15 km wide of the UFS, and featured by 0.44 mm/yr of averaged tectonic
strike-slip deformation between the Oligocene and the present day. The aforementioned
authors described different surface segments of the UFS of right-lateral strike-slip
ranging between 12 and 14 km length. The structural data collected by Tavani et al.
(2011) pointed out the 60% of data correspond to right lateral strike-slip with WNW-
ESE trend, together with conjugate reverse faulting with NE-SW, NW-SE and E-W
trend, and left-lateral strike-slip faults N-S oriented. They concluded that this scheme
could be related to a transpressional right-lateral tectonic system with a maximum
horizontal compression, $S_{Hmax}$, striking N120ºE. Concerning the geological evidence of



recent sediments affected by tectonic movements of the UFS, Tavani et al. (2011)
suggest Middle Miocene in time for this tectonic activity. However, geomorphic
markets (river and valley geomorphology) could indicate activity at present-times. All
of these data correspond to regional or small-scale data collected to explain the Basque-
Cantabrian Cenozoic transpressive basin. The advantage of the methodology proposed
here to establish the tectonic local regime affecting the reservoir, is the searching for
local-scale tectonics (20 km sized), and the estimation of the depth for the non-
deformation surface for strata folding in transpressional tectonics (Lisle et al., 2009).
(2) Regarding the stress field from earthquake focal mechanism solutions, Herraiz et al.
(2000) pointed out the regional trajectories of $S_{Hmax}$ with NNE-SSW trend, and with a
NE-SW $S_{Hmax}$ trend from slip-fault inversion data. Stich et al. (2006) obtained the stress
field from seismic moment tensor inversion and GPS data. These authors pointed out a
NW-SE Africa-Eurasia tectonic convergence at tectonic rate of 5 mm/yr approximately.
However, no focal mechanism solutions are found within the Hontomín area (20 km)
and only long-range spatial correlation could be made with high uncertainty (in time,
space and magnitude). The same lack of information appears in the work of De Vicente
et al. (2008), with no focal mechanism solutions in the 50 km surrounding the HPP. In
this work, these authors classified the study area as uniaxial extension to strike-slip with
NW-SE $S_{Hmax}$ trend.
Regional data about the tectonic field within the HPP (Bureba basin), inferred from
different works (Herraiz et al., 2000; Stich et al., 2006; De Vicente et al., 2008, 2011;
Tavani et al., 2011; Tavani, 2012), show differences for the $S_{Hmax}$ trend. These works
explain the tectonic framework for regional scale, nevertheless local tectonics could
determine the low permeability and the potential Induced Seismicity within the
Hontomín reservoir. In the next section, we have applied the methodology described at



the section 3 of this manuscript, in order to compare the regional results from these
works and to establish the tectonic evolution of the Burgalesa Platform.

*2.3 Strategy of the ENOS European Project*
Hontomín pilot-plant (HPP) for $CO_2$ onshore storage is the only one in Europe
recognized as a key-test-facility, and it is managed and conducted by CIUDEN
(*Fundación Ciudad de la Energía*). This HPP is located within the province of Burgos
(**Fig. 1b**), in the northern central part of Spain.
The methodology proposed in this work and its application for long-term onshore CCS
managing in the frame of geological risk, is based on the strain tensor calculation, as
part of the objectives proposed in the European project ENOS. The ENOS project is an
initiative of CO2GeoNet, the European Network of Excellence on the geological storage
of $CO_2$ for supporting onshore storage and fronting the associated troubles as CCS
perception, the safe storage operation, potential leaking management and health, and
environmental safety (Gastine et al., 2017). ENOS combines a multidisciplinary
European project, which focuses in onshore storage, with the demonstration of best
practices through pilot-scale projects in the case of Hontomín facilities. Moreover, this
project claims for creating a favorable environment for CCS onshore through public
engagement, knowledge sharing, and training (Gastine et al., 2017). In this context, the
work-package WP1 is devoted to "ensuring safe storage operations". The IGME team is
committed to develop and to carry out a technology to determine the active strain-field
affecting the sub-surface reservoir and fault patterns and to assign the fault type for the
estimation of potential fault-patterns as low-permeability paths as well.






3. METHODS AND RATIONALE
The lithosphere remains in a permanent state of deformation, related to plate tectonics
motion. Strain and stress fields are the consequence of this deformation on the upper
lithosphere, arranging different fault patterns that determine sedimentary basins and
geological formations. Kinematics of these faults describes the stress/strain fields, for
example measuring grooves and slickensides on fault planes (see Angelier, 1979,
Reches, 1983 among others). The relevance of the tectonic field is that stress and strain
determine the earthquake occurrence by the fault activity. In this work, we have
performed a brittle analysis of the fault kinematics, by measuring slickenfiber on fault
planes in several outcrops in the surroundings of the onshore reservoir. These faults
were active during the Mesozoic, and from Late Miocene to Quaternary. To carry out
the methodology proposed in this work, the study area was divided in a circle with four
equal areas, and we searched outcrops of fresh rock to perform the fault kinematic
analysis. This allows establishing a realistic tectonic very-near field to be considered
during the storage seismic monitoring and long-term management.

2.1 *Paleostrain Analysis*
We have applied the strain inversion technique to reconstruct the tectonic field
(paleostrain evolution), affecting the Hontomín site between the Triassic, Jurassic,
Cretaceous and Neogene ages (late Miocene to present times). For a further
methodology explanation see Etchecopar et al. (1981), Reches (1983) and Angelier
(1990). The main assumption for the inversion technique of fault population is the self-
similarity to the scale invariance for the stress/strain tensors. This means that we can
calculate the whole stress/strain fields by using the slip data on fault planes and for
homogeneous tectonic frameworks. The strain tensor is an ellipsoid defined by the



orientation of the three principal axes and the shape of the ellipsoid (k). This method
assumes that the slip-vectors, obtained from the pitch of the striation on different fault
planes, define a common strain tensor or a set in a homogeneous tectonic arrangement.
We assume that the strain field is homogeneous in space and time, the number of faults
activated is greater than five and the slip vector is parallel to the maximum shear stress
($\tau$).
The inversion technique is based on the Bott equations (Bott, 1959). These equations
show the relationship between the orientation and the shape of the stress ellipsoid:

Tan ($\theta$) = [n / (l * m)] * [$m^2$ - (1- $n^2$) * R']           [eq.1]
R' = ($\sigma_z$ - $\sigma_x$) / ($\sigma_y$ - $\sigma_x$)                               [eq.2]

Where l, m and n are the direction cosines of the normal to the fault plane, $\theta$ is the pitch
of the striation and R' is the shape of the stress ellipsoid obtained in an orthonormal
coordinate system, x, y, z. In this system, $\sigma_y$ is the maximum horizontal stress, $\sigma_x$ is the
minimum horizontal stress axis and $\sigma_z$ is the vertical stress axis.

*3.2 The Right-Dihedral Model for Paleostrain Analysis*
The Right-Dihedral (RD) is a semi-quantitative method based on the overlapping of
compressional and extensional zones by using a stereographic plot. The final plot is an
interferogram figure which usually defined the strain-regime. This method is strongly
robust for conjugate fault sets and with different dip values for a same tensor. The RD
was originally defined by Pegoraro (1972) and Angelier and Mechler (1977), as a
geometric method, adjusting the measured fault-slip data (slickensides) in agreement
with theoretical models for extension and compressive fault-slip. Therefore, we can




constraint the regions of maximum compression and extension related to the strain
regime.

*3.3 The Slip Model for the Paleostrain Analysis*
The Slip Model (SM) is based on the Navier-Coulomb fracturing criteria (Reches,
1983), taking the Anderson model solution for this study (Anderson, 1951; Simpson,
1997). The Anderson model represents the geometry of the fault plane as monoclinic,
relating the quantitative parameters of the shape parameter (K') with the internal
frictional angle for rock mechanics ($\phi$) (De Vicente 1988, Capote et al., 1991).
Moreover, this model is valid for neoformed faults, and some considerations have to be
accounted for previous faults and weakness planes present in the rock. These
considerations are related to the dip of normal and compressional faults, such as for
compressional faulting values lower than b < 45°, reactivated as extensional faults. This
model shows the relationships between the K', $\phi$ and the direction cosines for the
striation on the fault plane (De Vicente, 1988; Capote et al., 1991):

$K' = e_y / e_z$                                      [eq.3]

Where $e_z$ is the vertical strain axis, $e_y$ is the maximum horizontal shortening and $e_x$ is
the minimum horizontal shortening. This model assumes that there is no change of
volume during the deformation and $e_y = e_x + e_z$.
For isotropic solids, principal strain axes coincide with the principal stress axes. This
means that in this work, the orientation of the principal stress axis, $S_{Hmax}$ is parallel to
the orientation of the principal strain axes, $e_y$, and hence, the minimum stress axis, $S_{hmin}$,



is parallel to the minimum strain axis, $e_x$. This assumption allows us to estimate the
stress trajectories ($S_{Hmax}$ and $S_{hmin}$) from the $De_y$ SM results.
Resolving the equations of Anderson (Anderson, 1951) for different values, we can
classify the tectonic regime that activates one fault from the measurement of the fault
dip, sense of dip and pitch of the slickenside, assuming that one of the principal axes
($e_x$, $e_y$ or $e_z$) are vertical (Angelier, 1984). We can classify of the tectonic regime and
represent the strain tensor by using the $e_y$ and $e_x$ orientation.

*3.4 The K' strain diagram*
Another analysis can be achieved by using the K'-strain diagram developed by Kaverina
et al. (1996) and codified in python-code by Álvarez-Gómez (2014). These authors have
developed a triangular representation based on the fault-slip, where tectonic patterns can
be discriminated between strike-slip and dip-slip types. This diagram is divided in 7
different zones according to the type of fault: (1) pure normal, (2) pure reverse and (3)
pure strike-slip, combined with the possibility of oblique faults: (4) reverse strike-slip
and (5) strike-slip with reverse component, and lateral faults: (6) normal strike-slip and
(7) strike-slip faults with normal component (**Fig. 3**). Strike-slip faults are defined by
small values for pitch (p < 25º) and dips close to vertical planes (β > 75º). High pitch
values (p > 60º) are related to normal or/reverse fault-slip vectors. Extensional faults
show $e_y$ in vertical whereas compressional faults show $e_y$ in horizontal plane.
This method was originally performed for earthquake focal mechanism solutions by
using the focal parameters, the nodal planes (dip and strike) and rake. The triangular
graph is based on the equal-areal representation of the T, N or B and P axes in spherical
coordinates (T tensile, N or B neutral and P pressure axes), and the orthogonal
regression between Ms and mb for the Harvard earthquake CMT global catalogue in



1996. Álvarez-Gómez (2014) presented a code python-based for computing the
Kaverina diagrams, and we have modified the input parameters by including the K'
intervals for the strain field from the SM. The relationship between the original diagram
of Kaverina (**Fig. 3a**) and the K'-dip diagram (**Fig. 3b**) that we have used in this work is
shown in the figure 3. The advantage of this diagram is the fast assignation of the type
of fault and the tectonic regime that determine this fault pattern, and the strain axes
relationship.

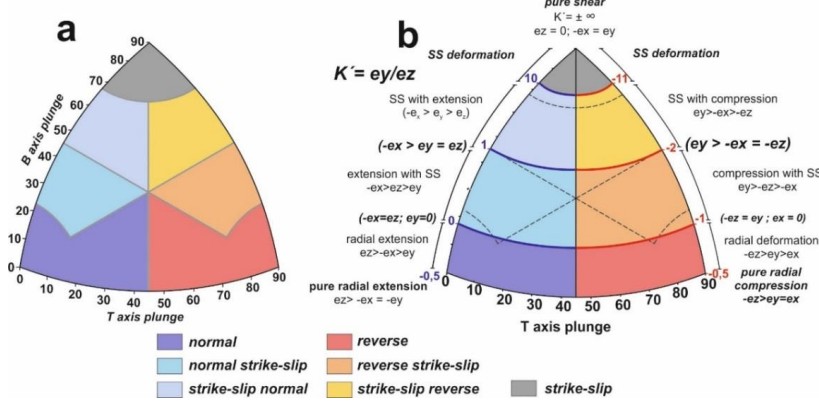


*Figure 3. a) Kaverina original diagram to represent the tectonic regime from an earthquake focal*
*mechanism population (see Kaverina et al., 1996 and Álvarez-Gómez, 2014). b) K'-strain diagram used*
*in this work. Dotted lines represent the original Kaverina limits. Colored zones represent the type of*
*fault. The tectonic regime is also indicated by the relationship between the strain axes and the colored*
*legend. SS Strike slip. The B axis is the orthogonal to the P and T axes.*

Table 1 summarizes the different tectonic regimes of the figure 3b showing the
relationship with the strain main axes $e_y$, $e_x$ and $e_z$. This diagram exhibits a great
advantage to classify the type of fault according to the strain tensor. Therefore, we can
assume the type of fault from the fault orientation affecting geological deposits for each
strain tensor obtained.





| K' | T-axis | strain axis rel. | fault type | tectonic field |
|---|---|---|---|---|
| < - 0.5 | 0º | $e_z>-e_x=-e_y$ | normal | pure radial extension |
| -0.5<K'<0 | 0º-45º | $e_z>-e_x>-e_y$ | normal | radial extension |
| K'=0 | 0º-45º | $e_z=-e_x; e_y=0$ | normal | plain strain |
| 0<K'<1 | 0º-45º | $-e_x>e_z> e_y$ | normal with SS | extension with shear |
| k=1 | 0º-45º | $-e_x>e_y=e_z$ | normal with SS | extension with shear |
| 1<K'<10 | 0º-45º | $-e_x>e_y>e_z$ | strike-slip with N | shear with extensional |
| 10<K'<∞ | 0º-45º | ------------ | strike-slip | shear deformation |
| K'=∞ | 45º | $e_z=0;-e_x=e_y$ | strike-slip | pure shear deformation |
| ∞<K'<-11 | 45º-90º | ------------ | strike-slip | shear deformation |
| -11<K'<-2 | 45º-90º | $e_y>-e_x>-e_z$ | strike-slip with R | shear with compression |
| K'=-2 | 45º-90º | $e_y>-e_x=-e_z$ | reverse with SS | compression with shear |
| -2<K'<-1 | 45º-90º | $e_y>-e_z>-e_x$ | reverse with SS | compression with shear |
| K'=-1 | 45º-90º | $-e_z=e_y; e_x=0$ | reverse | plain strain |
| -1<K'<-0.5 | 45º-90º | $-e_z>e_y>e_x$ | reverse | radial compression |
| K'=-0.5 | 45º-90º | $-e_z>e_y=e_x$ | reverse | pure radial compression |

SS = strike-slip          $e_x$ = value of the minimum horizontal shortening

N= normal                 $e_y$ = value of the maximum horizontal shortening

R= reverse                $e_z$ = value of the vertical axis


*Table 1. Different tectonic regimes, K' values, dip values and fault type for the Kaverina modified*
*diagram used in this work. According to the strain axes relationship, faults can be classified and the*
*tectonic regime can be established.*

*3.5 The Circular-Quadrant-Search (CQS) strategy for the paleostrain analysis*
In this work, we propose a low-cost strategy based on a well-known methodology for
determining the stress/strain tensor affecting a CCS reservoir, which will allow for
monitoring long-term geological and seismic behavior. The objective is to obtain
enough structural data and spatially homogeneous of faults (**Figs. 4, 5**), for
reconstructing the stress/strain tensor by using the methodologies described above. The
key-point is the determination of the orientation of the $e_y$, $e_x$ and K' to plot in a map and
therefore, to establish the tectonic regime. We have chosen quadrants of the circles with
the aim to obtain a high-quality spatial distribution of point for the interpretation of the

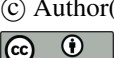


local and very near strain field. Hence, data are homogeneously distributed, instead of
being only concentrated in one quadrant of the circle.

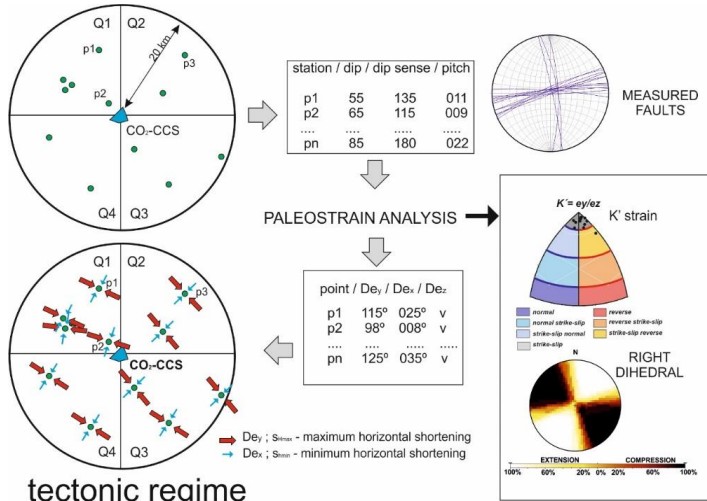

*Figure 4. Methodology proposed to obtain the strain field affecting the CCS reservoir. The distances for*
*outcrops and quadrants proposed is 20 km. The technique of Right Dihedral and the K' strain diagram is*
*described in the main text. The $e_y$ and $e_x$ represented are a model for explaining the methodology. $De_y$*
*and $De_x$ are the direction of the maximum and minimum strain, respectively. Blue box at the center is the*
*$CO_2$ storage geological underground formation.*

Pérez-López et al. (2018) carried out a first approach to the application of this
methodology at Hontomín, under the objective of the ENOS project (see next section
for further details). We propose a circular searching of structural field stations (**Figs. 4,**
**5**), located within a 20 km radius. This circle was taken given that active faults with the
capacity of triggering earthquakes of magnitudes close to M 6, exhibits a surface rupture
of tens of kilometers, according to the empirical models (Wells and Coppersmith,
1994). Moreover, Verdon et al. (2015) pointed out that the maximum distance of
induced earthquakes for fluid injection is 20 km. Larger distances could not be related
to the stress/strain regime within the reservoir, except for the case of large geological

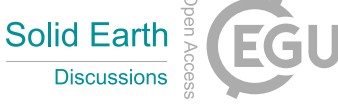



structures (folds, master faults, etc.). Microseismicity in CCS reservoir is mainly related
to the operations during the injection/depletion stages and long-term storage (Verdon
2014, Verdon et al., 2015, McNamara, 2016).

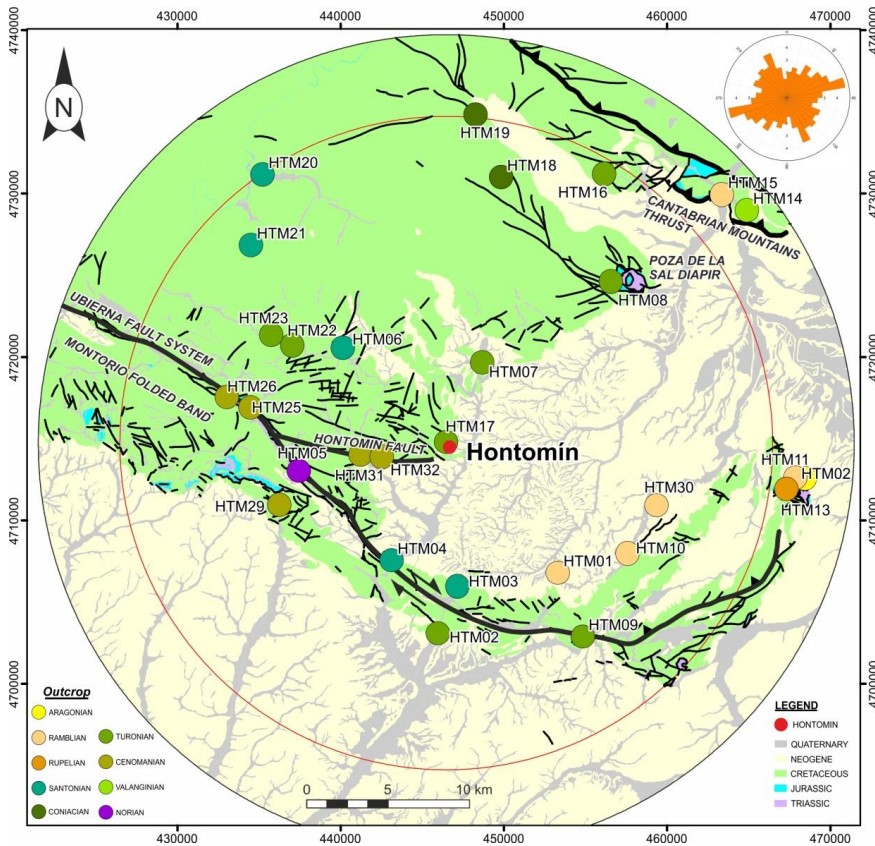


*Figure 5. Geographical location of field outcrops in the eastern part of the Burgalesa Platform domain.*
*Black lines: observed faults; red circle: 20km radius study zone. A total of 447 fault data were collected*
*in 32 outcrops. Data were measured by a tectonic compass on fault planes at outcrops. The spatial*
*distribution of the field stations is constrained by the lithology. Coordinates are in meters, UTM H30.*

The presence of master faults (capable to trigger earthquakes of magnitude = or > than 6
and 5 km long segment) inside the 20 km radius circle, implicates that the regional
tectonic field is relevant for the reservoir geodynamics, being responsible for the strain
accumulation in kilometric fault-sized. Furthermore, the presence of master faults could





increase the occurrence of micro-earthquakes, due to the presence of secondary faults
prone to trigger earthquakes by their normal seismic cycle. Bearing in mind that CCS
onshore reservoirs use to be deep saline aquifers (e.g. Bentham and Kirby, 2005), as
Hontomín is (Gastine et al., 2017, Le Gallo and de Dios, 2018), and be related to
folding and fractured deep geological structures, local tectonics plays a key role in
micro-seismicity and the possibility of $CO_2$ leakage.
The constraints of this strategy are related to the absence of kinematics indicators on
fault planes, due to the geomechanical property of the lithology involved or the erase by
later geological processes as neoformed mineralization, etc. A poor spatial distribution
of the outcrops was also taken into account for constraining the strategy. The age of
sediments does not represent the age of the active deformations and hence, the active
deformation has to be analyzed by performing alternative methods (i.e.
paleoseismology, archaeoseismology).

4. RESULTS
*4.1 Strain Field Analysis*
We have collected 447 fault-slip data on fault planes in 32 outcrops, located within a 20
km radius circle centered at the HPP (**Fig. 5**). The age of the structural field stations
ranges between Early Triassic to present-day and are mainly located in Cretaceous
limestones and dolostones (**Fig. 5, Table 2**). No Jurassic outcrops were located, and
seven stations are located on Neogene sediments, ranging between Early Oligocene to
Middle-Late Miocene. The short number of Neogene stations is due to the mechanical
properties of the affected sediments, mainly poor-lithified marls and soft-detrital fluvial
deposits. Despite that, all the Neogene stations exhibit high-quality data with a number
of fault-slip data ranging between 7 and 8, enough for a minimum quality analysis.



We have labeled the outcrops with the acronym HTM followed by a number (see **figure**
**5** for the geographical location and **Table 2** and **figure 6** for the fault data). The station
with the highest number of faults measured is HTM17 with 107 faults on Cretaceous
limestone. Nevertheless, we have removed the HTM17 to the analysis due to the high
number of measurements, including lot of noise that could disturb the whole analysis.
Conjugate fault systems can be recognized in most of the stations (HTM1, 3, 5, 7, 10,
14, 16, 21, 23, 25, 26, 29, 30 and 32, **Fig. 6**), although there are a few stations with only
one well defined fault set (6, 22, 32). We have to bear in mind that recording of
conjugate fault systems are more robust for the brittle analysis than recording isolated
fault sets, better constraining the solution (Žaholar and Vrabec, 2008). In total, 29 of 32
stations were used (HTM24, 27, 28 with no quality data), and from these 29 stations, 24
were analyzed with the paleostrain technique. Solutions obtained here are robust to
establish the strain field as the orientation of the $e_y$, $S_{Hmax}$ (**Figs. 7** and **8**).





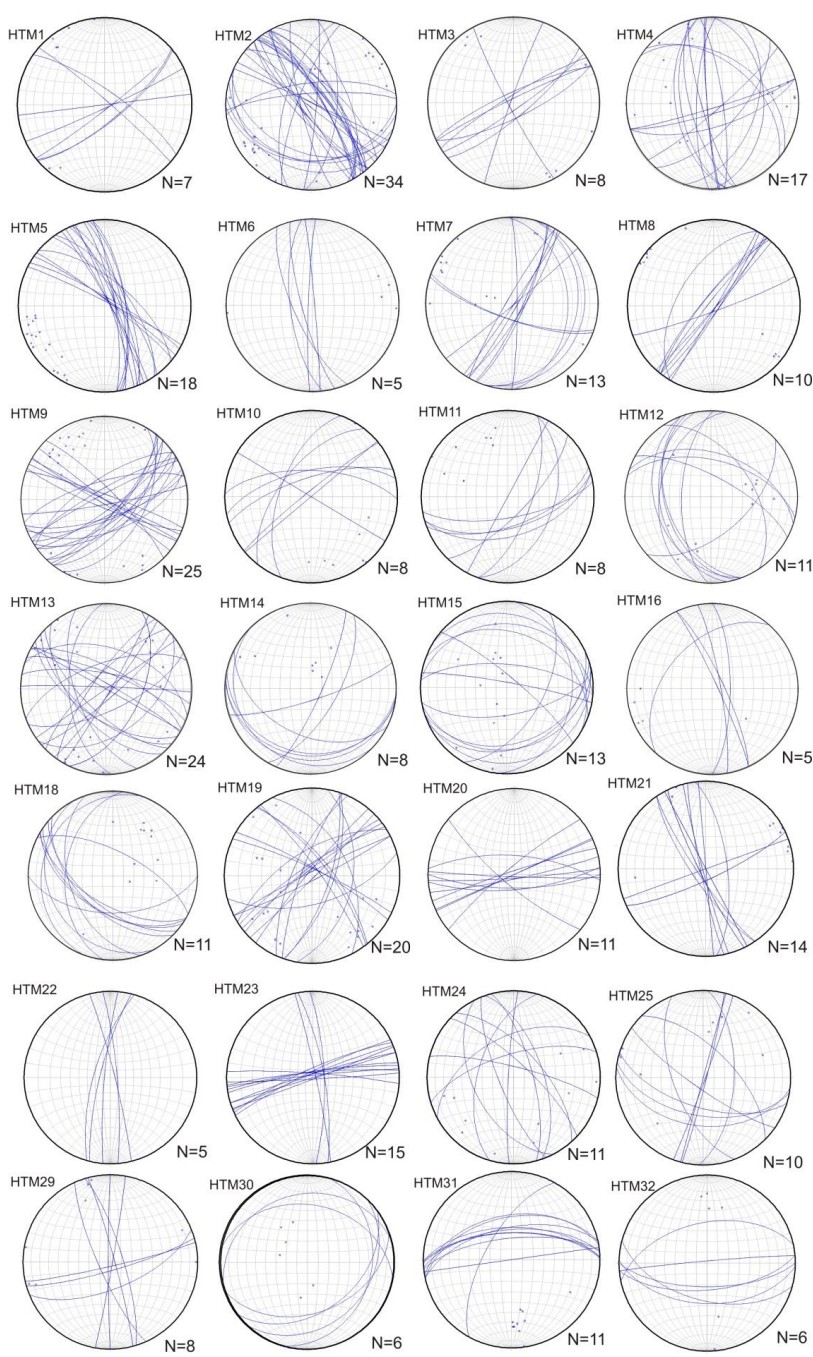


*Figure 6. Stereographic representation (cyclographic plot in Schmidt net, lower hemisphere) of the fault*
*planes measured in the field stations. "n" is the number of available data for each geoestructural station.*
*HTM24, 27, 28 are not included due to lack of data, and HTM17 due to the high number of faults.*






The results obtained from the application of the paleostrain method have been expressed
in stereogram, right dihedral (RD), slip method (SM) and K'- diagram (**Fig. 7**). The K'-
diagram shows the fault classification as normal faults, normal with strike-slip
component, pure strike-slip, strike-slip with reverse component and reverse faults (see
**Fig. 3**). Main faults are lateral strike-slips and normal faults, followed by reverse faults,
strike-slips and oblique strike-slips faults. The results of the strain regime are as
follows: 1) 43% of extensional with shear component; 2) 22% of shear; 3) 13% of
compressive strain (lower Cretaceous and early-middle Miocene, **Table 2**); 4)13% of
pure shear and 5) 9% of shear with compression strain field, although with the presence
of five reverse faults.
On the other hand, we can observe that there are solutions with a double value for the
$e_y$, $S_{Hmax}$ orientation: HTM1, 2, 10, 11, 13, 15, 19, 26, and 30. The stations HTM3 and
23 (upper Cretaceous), show the best solution for strike-slip strain field as a pure strike-
slip regime and $e_y$ with N25°E and N99ºE trend, respectively (**Fig. 7**).
It is easy to observe the agreement between the $e_y$ results from the SM and the K'- strain
diagram, for instance, in the HTM 2 the K'-diagram indicates strike-slip faults with
reverse component for low dips (0º < β < 40º) but also indicates strike-slip faults with
normal component for larger dips (40º < β < 90º). However, both results are in
agreement with a strain field defined by the orientation for $e_y$, $S_{Hmax}$ with N150º ± 18°
trend. This tectonic field affects Cretaceous carbonates and coincides with the regional
tectonic field proposed by Herraiz et al. (2000), Stich et al. (2006),Tavani et al. (2011)
and Alcalde et al. (2014).





**FIGURE 7a**





**FIGURE 7b**





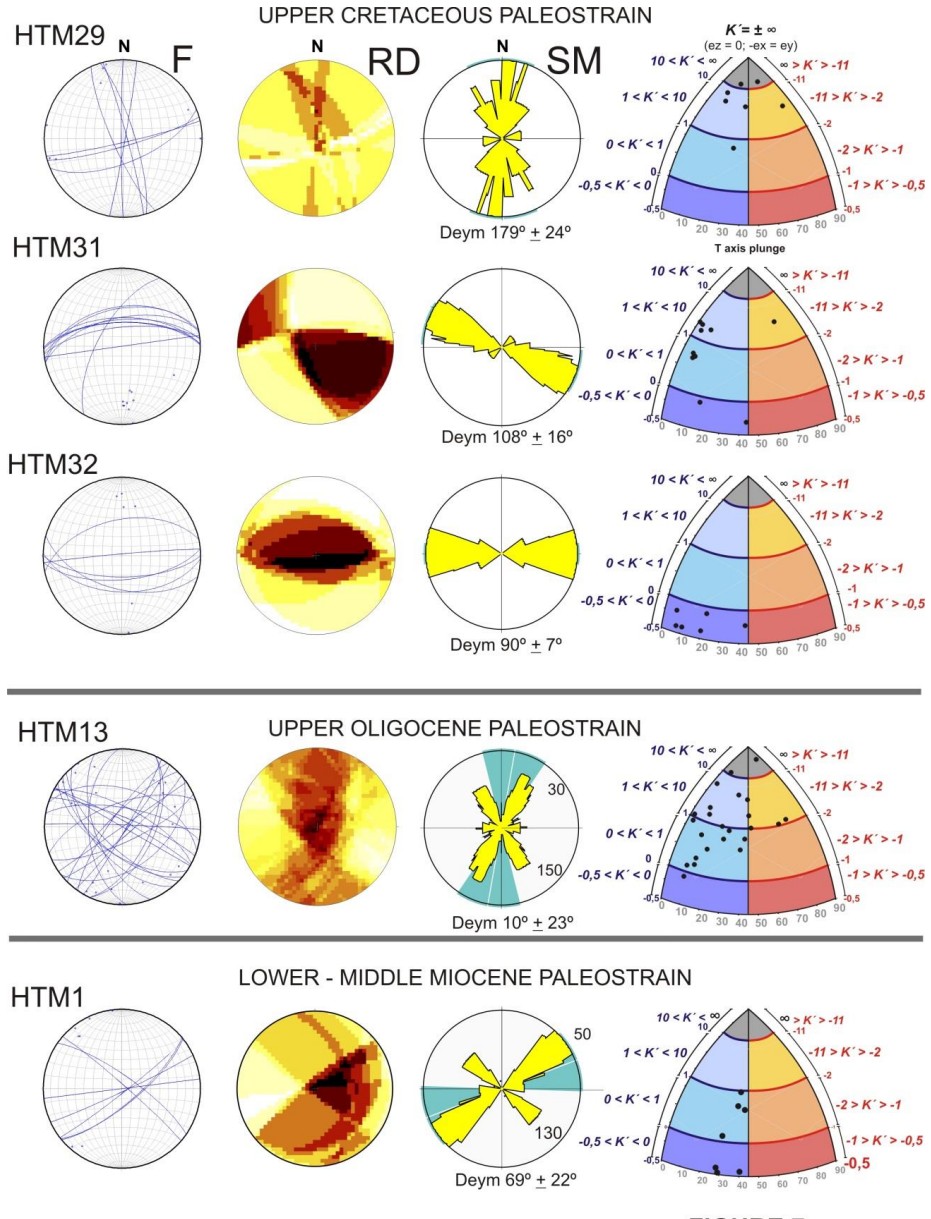

491          **FIGURE 7c**





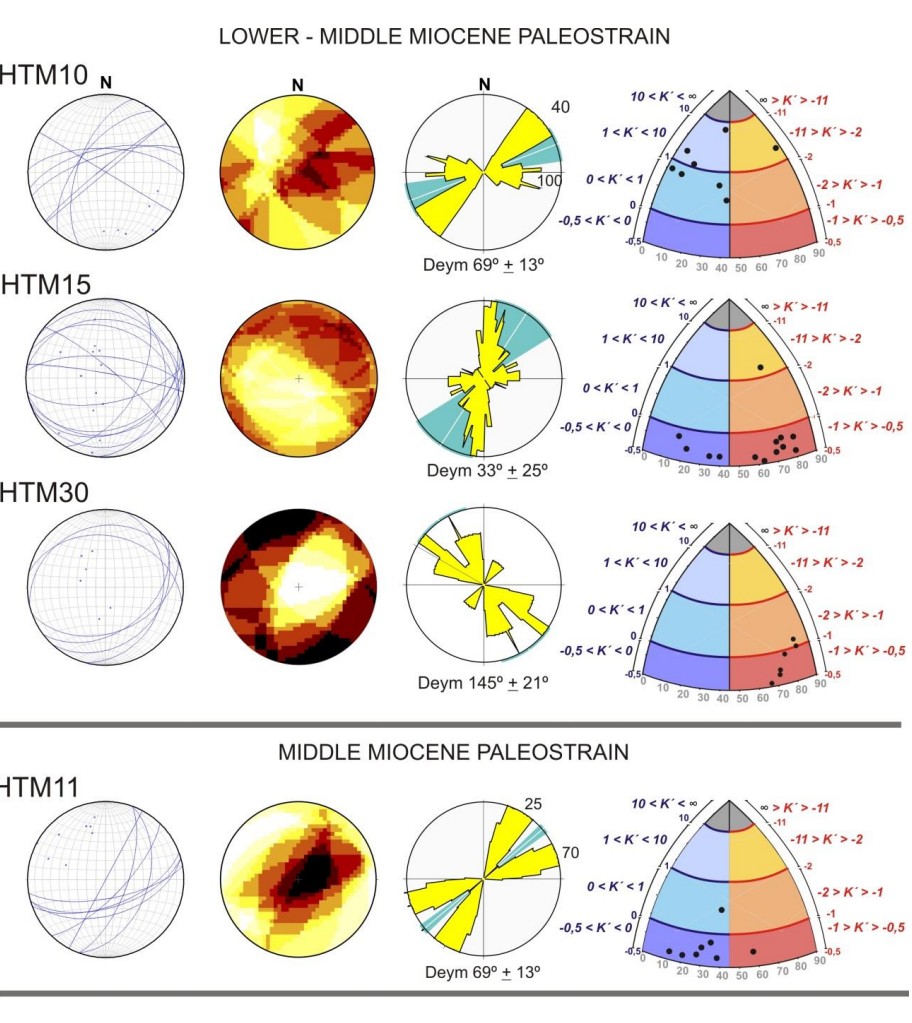

FIGURE 7d

Figure 7. Results of the paleostrain analysis obtained and classified by age. Deym: striking of the averaged of the $De_y$ value; F: fault stereographic representation; K': diagram with dots for each fault slip solution; RD: Right Dihedral method; SM: Slip Method, K'. See Methods for further explanation.



| STATION | nº faults | series/epoch | Dey main direction | dispersion | strain tensor |
|---|---|---|---|---|---|
| HTM05 | 18 | UPPER TRIASSIC | 140 | 8 | NORMAL STRIKE-SLIP |
| HTM14 | 8 | LOWER CRETACEOUS | 34 | 21 | COMPRESSION |
| HTM02 | 34 | UPPER CRETACEOUS | 150 | 18 | STRIKE-SLIP (N-C) |
| HTM03 | 8 | UPPER CRETACEOUS | 25 | 6 | STRIKE-SLIP (N-C) |
| HTM08 | 10 | UPPER CRETACEOUS | 45 | 11 | STRIKE-SLIP |
| HTM17 | 105 | UPPER CRETACEOUS | 107 | 24 | NORMAL |
| HTM19 | 20 | UPPER CRETACEOUS | 61 | 30 | NORMAL STRIKE-SLIP |
| HTM20 | 11 | UPPER CRETACEOUS | 75 | 5 | STRIKE-SLIP |
| HTM21 | 14 | UPPER CRETACEOUS | 138 | 22 | STRIKE-SLIP NORMAL |
| HTM22 | 5 | UPPER CRETACEOUS | 175 | 8 | NORMAL STRIKE-SLIP |
| HTM23 | 14 | UPPER CRETACEOUS | 99 | 15 | STRIKE-SLIP |
| HTM25 | 11 | UPPER CRETACEOUS | 141 | 26 | STRIKE-SLIP COMPRESSION |
| HTM26 | 10 | UPPER CRETACEOUS | 0 | 23 | STRIKE-SLIP |
| HTM29 | 8 | UPPER CRETACEOUS | 179 | 24 | STRIKE-SLIP NORMAL |
| HTM31 | 11 | UPPER CRETACEOUS | 108 | 16 | STRIKE-SLIP NORMAL |
| HTM32 | 6 | UPPER CRETACEOUS | 90 | 7 | NORMAL |
| HTM13 | 24 | EARLY OLIGOCENE | 25-160 | 23 | NORMAL STRIKE-SLIP |
| HTM01 | 7 | EARLY-MIDDLE MIOCENE | 70 | 22 | NORMAL STRIKE-SLIP |
| HTM10 | 8 | EARLY-MIDDLE MIOCENE | 69 | 13 | NORMAL STRIKE-SLIP |
| HTM15 | 13 | EARLY-MIDDLE MIOCENE | 33 | 25 | COMPRESSION |
| HTM30 | 6 | EARLY-MIDDLE MIOCENE | 145 | 21 | COMPRESSION |
| HTM11 | 8 | MIDDLE MIOCENE | 50 | 4 | NORMAL STRIKE-SLIP |


*Table 2. Summary of the outcrops showing the number of faults, the type of the strain tensor obtained, the*
*$De_y$, $S_{Hmax}$ striking and the age of the affected geological materials. Asterisk indicates those field stations*
*detailed in the figure 7. N-C is normal component for strike-slip movement.*

Two $e_y$, $S_{Hmax}$ directions can be considered, N150ºE and N50ºE. We obtained an
averaged value of N105ºE by mixing both values of trend. However, a large number of
measured faults and their uncertainties slightly disturb the results (**Table 2**).



*Figure 8) Upper left frame: Synthesis of the K'- map obtained from Giner-Robles et al. (2018) for the*

*whole Iberian Peninsula from focal mechanism solutions. HPP is located between a triple junction of K'*

*defined by compression towards the north, extension to the southeast and strike-slip to the west. Black*

*dots are the earthquakes with focal mechanism solutions used. Sketches represent the $e_y$, $S_{Hmax}$*

*trajectories obtained from the outcrops for early Triassic, Early Cretaceous, Late Cretaceous, Early*





*Oligocene, Early – Middle Miocene and Present-day strain field from Herraiz et al. (2000). Main*
*structures activated under the strain field defined are also included.*

*4.2 Late Triassic Paleostrain*
Strain analysis from HTM5 fault set shows $e_y$ with NW-SE trending and shear regime
with extension defined by strike-slip faults (**Figs. 7a** and **8**). This is in agreement with
the uniaxial extension described in Tavani (2012), author that constrain this regime with
$S_{hmin}$ with NE-SW trending.

*4.3 Cretaceous Paleostrain*
HTM 14 is the only outcrop from early Cretaceous age, showing a compressive tectonic
stage with reverse fault solutions, defined by $e_y$ with NE-SW trending (**Fig. 8**). Taking
into account the extensional stage related to the Main Rifting Stage (i.e. Carola, 2004;
Tavani, 2012; Tugend et al., 2014) during this age, we interpreted these results as a
modern strain field, probably related to the Cenozoic Inversion stage. A local
compressive stage was discarded due to the absence of compressive structures related to
this age in the area and surroundings.
Outcrops HTM 2, 17, 19, 20, 21, 23, 25, 26, 29, 31 and 32 are from the upper
Cretaceous carbonates, and four main strain fields are described, depending on the fault
sets (**Fig. 8**): (1) a compressive stage featured by $e_y$ with NW-SE trending, similar to
those stage described in Tavani (2012), (2) a normal strain stage with $e_y$ striking both E-
W and NE-SW (**Fig. 8**, HTM 20, 21, 31 and 32). Finally, a (3) a shear stage (activated
strike-slip faults) and (4) a shear with an extension (strike-slip with normal component)
were described as well. These two late stages are featured by $e_y$ with NE-SW and NW-
SE trends. The existence of four different strain fields is determined by different ages
during the Cretaceous and different spatial locations in relation to the main structures,



the Ubierna Fault System, Hontomin Fault, Cantabrian Thrust, Montorio folded band
and the anticline (**Fig. 8**).

*4.4 Cenozoic strain field*
The Cenozoic tectonic inversion was widely described in the area by different authors
(e.g. Carola, 2004; Tavani, 2012; Tungend et al., 2014). This tectonic inversion is
related to compressive structures, activating NW-SE and NE-SW thrusts with NW-SE
and NNE-SSW $e_y$ trends, respectively. The Ubierna Fault has been inverted with a
right-lateral transpressive kinematics during the Cenozoic (Tavani et al., 2011). Early
Oligocene outcrop (HTM13, **Figs. 7c** and **8**) shows an extensional field with $e_y$ with
NNE-SSW trending. During the Miocene, HTM15 and HTM30 outcrops exhibit the
same $e_y$ striking, but for compressive tectonic stage (**Figs. 7c** and **8**). In addition, during
the middle Miocene, an extensional tectonic strain is described and characterized by
NNE-SSW and NE-SW trends. Summarizing, the Cenozoic inversion and tectonic
compression are detected during the early middle Miocene, later to the Oligocene, but
during the middle Miocene, only one extensional stage was interpreted.
The outcrops located closer to the HPP (HTM 17, 31, 32, **Figs. 5 and 7**), show E-W
faults. HTM05 is located on the Ubierna Fault, showing NW-SE, whilst HTM03 shows
strike-slip NE-SW. Moreover, this station is located within a valley with the same
orientation and surrounding faults have the same orientation (**Fig. 5**). Close to the HPP
facilities, E-W faults are measured. This fault set was activated under a strain field
defined by $e_y$ with E-W trending and K'-diagram shows normal faults with strike-slip
component. However, the present-day $e_y$ with a roughly N-S trend (Herraiz et al., 2000,
Stich et al., 2006), could active E-W faults as reverse faults and hence, more energy





would needed to move like seismic sources. In addition, fault dip data obtained from
structural analysis can be included in geomechanical analysis of fault rupture.
Strain analysis suggests that the planes that could affect the leakage into the reservoir
would be those planes parallel to the $S_{Hmax}$ orientation, that is, NNW-SSE and N-S (**Fig.**
**7**). Moreover, N50ºE $S_{Hmax}$ orientation could also affect the reservoir. HPP facilities are
close to WNW-ESE fault although the HTM17 station shows that N-S fault planes
could play an important role for seepage fluid into the reservoir.

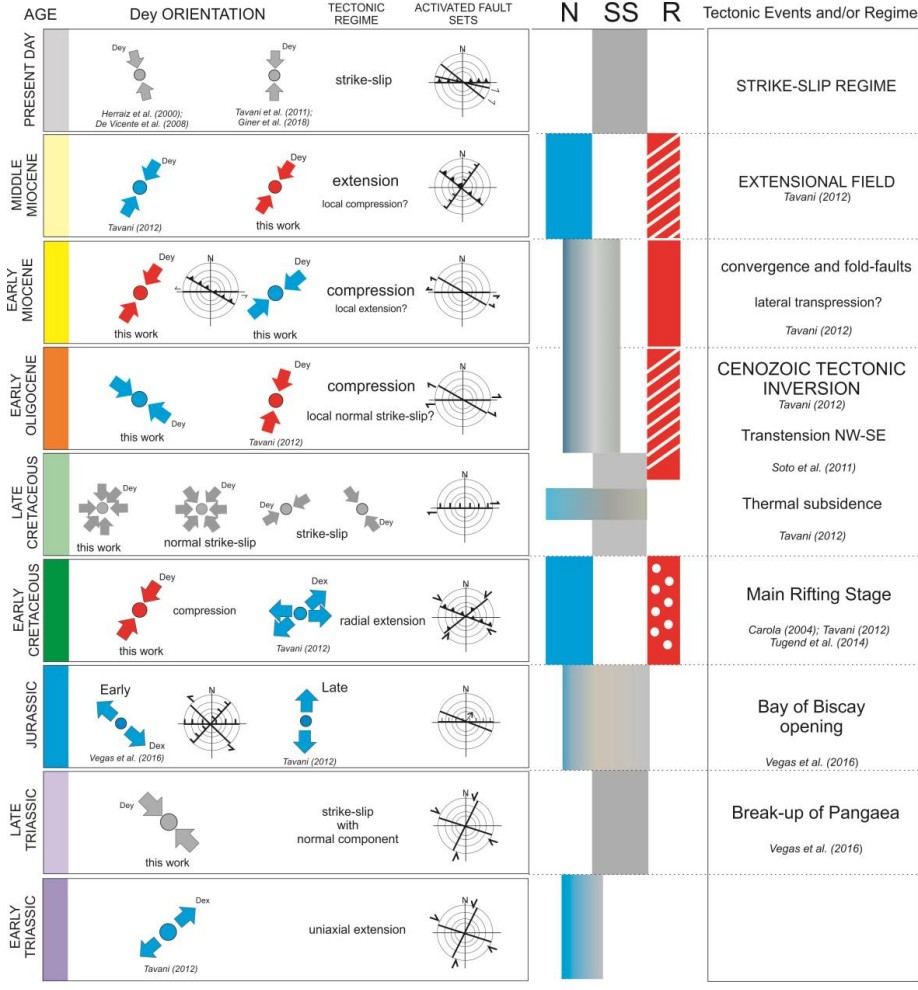




*Figure 9. Tectonic field evolution of the Burgalesa Platform domain (20-km radius circle centered at*
*HPP), interpreted from the paleostrain analysis. The regional tectonic field from other authors are also*
*included. Ages were estimated from the HTM outcrops affecting geological well-dated deposits. N:*
*extensional faulting; R: compressive faulting; SS: strike-slip faulting. Red and white means local*
*paleostrain field, and red with white dots means superposed modern paleostrain field.*

5. DISCUSSION
*5.1. Mesozoic – Cenozoic Paleostrain evolution*
We propose a tectonic field evolution for the Hontomín CCS area, from the Triassic to
Neogene times (**Fig. 9**), based on the results obtained from the paleostrain analysis (**Fig.**
**8**). Furthermore, the data used in this work were completed from available bibliographic
data of paleostress and paleostrain affectingn the Burgalesa Platform (see references in
**Fig. 9**) and large-scale tectonic events.
Triassic age is featured by a uniaxial extension determined by a paleostrain field with $e_x$
striking NE-SW (Tavani, 2012). The oldest tectonic strain field that we have obtained,
recorded during the Late Triassic, is represented by a strike-slip tectonic field with shear
component (**Figs 7a, 8** and **9**), which we have related with the break-up of Pangaea. In
this stage, NE-SW right lateral faults are dominant.
No Jurassic outcrops are in the studied area and hence, the Jurassic deformation
assumed in this work comes from Tavani (2012), suggesting the aperture of the Bay of
Biscay in a large-scale N-S extension. Alcala et al. (2014) pointed out a tectonic
evolution from Lias diparism (Early Jurassic) and N-S tectonic extension, activating E-
W extensional faults. Moreover, Vegas et al. (2016) interpreted a rift extension during
this period.
The Early Cretaceous tectonic field shows a $e_y$ with NE-SW trend, determining a
compressive and convergence local stage (**Figs 7a, 8**). However, we have assumed this



strain field as modern, probably during the Cenozoic inversion, overlapping the
extensional regional paleostrain (**Fig. 9**, red with white circles). The Upper Cretaceous
tectonic strain is defined by several $e_y$ trends, from an initial N-S and NE-SW, to NW-
SE in a final transtensional state, which is in agreement with Soto et al. (2011), being
active even during the Early Oligocene.
The tectonic convergence represented by a NE- trending $S_{Hmax}$, determining a reverse
field to have taken place during the Early Miocene, although Tavani (2012) pointed out
that the Cenozoic tectonic inversion could start at the Upper Cretaceous. During the
Miocene, normal faults with shear during this period (**Fig. 9**) could be interpreted as
folding fractures. In this case, extensional faulting could appear in the upper part of
anticlines formed by bending. The middle Miocene is interpreted as an extensional stage
with normal strain field.
Finally, the active strain field (Miocene-Present-day), shows a local compressional field
with N50ºE trending $e_y$, $S_{Hmax}$ with and the regional field with N150ºE trending $e_y$,
$S_{Hmax}$. The active regional field proposed by Herraiz et al. (2000), Stich et al. (2006),
Tavani et al. (2011) and Alcalde et al. (2014), shows $e_y$, $S_{Hmax}$ with almost NNW-SSE
and N-S trends.

*5.2 Active faulting in the surrounding of HPP*
Quaternary tectonic markers for the UFS are suggested by Tavani et al. (2011).
According to the tectonic behavior of this fault as right-lateral strike-slip, and the fault
segments proposed by Tavani et al. (2011), ranging between 12 and 14 km long, the
question is whether this fault could trigger significant earthquakes and which could be
the maximum associated magnitude. This is a relevant question given that the "natural
seismicity" in the vicinity could affect the integrity of the caprock. Bearing in mind the



expectable long-life for the reservoir, estimated in thousands of years, the potential
natural earthquake that this master fault could trigger has to be estimated. In this sense,
it is necessary to depict seismic scenarios related to large earthquake triggering;
however, this type of analysis is beyond the focus of this work.
The income information that we have to manage in the area of influence (20 km) is: (a)
the instrumental seismicity, (b) the geometry of the fault, (c) the total surface rupture,
(d) the upper crust thickness and (e) the heat flow across the lithosphere. Starting for the
heat flow value, the Hontomín wells show a value that lies between 62 and 78 mW/m$^2$
at a 1,500 m depth approximately (Fernández et al., 1998). Regarding the Moho depth
in the area, these aforementioned authors obtained a value ranging between 36 and 40
km depth, while the lithosphere ranges between 120 and 130 km depth (Torne et al.,
2015). The relevance of this value is the study of the thermal weakness into the
lithosphere that could nucleate earthquakes in intraplate areas (Holford et al., 2011). For
these authors, the comparison between the crustal heat-flow in particular zones, in
contrast with the background regional value, could explain large seismicity and high
rates of small earthquakes occurrence, as the case of the New Madrid seismic zone. For
example, in Australia heat-flow values as much as 90 mW/m$^2$ are related with
earthquakes M > 5.
Regarding the maximum expected earthquake into the zone, we have applied the
empirical relationships obtained by Wells and Coppersmith (1994). We have used the
equations for strike-slip earthquakes according to the strain field obtained in the area
(pure shear), and the surface rupture segment for the Ubierna Fault System, assuming a
surface rupture segments between 12 and 14 km (Tavani et al., 2011). The obtained
results show that the maximum expected earthquake ranges between M 6.0 and M 6.1.
Wells and Coppersmith (1994) indicate for these fault parameters a total area rupture





ranging between 140 and 150 km$^2$. Surface fault traces rupture as lower as 7 km needs
at least 20 km of depth in order to reach a value of the fault-area rupturing greater than
100 km$^2$, in line with a Moho between 36 and 40 in depth.
Regarding the instrumental earthquakes recorded into the area, the two largest
earthquakes recorded correspond to magnitude M 3.4 and M 3.3, with a depth ranging
between 8 and 11 km, respectively, and a felt macroseismic intensity of III (EMS98,
www.ign.es, last access on May, 2019).Both earthquakes occurred between 50 and 60
km of distance from the Hontomín Pilot Plant. Only five earthquakes have been
recorded within the 20-km radius area of influence and with small magnitudes ranging
between M 1.5 and M 2.3. The interesting data is the depth of these earthquakes,
ranging between 10 and 20 km, which suggest that the seismogenic crust could reach 20
km of depth.
Furthermore, if E-W sets act as extensional faults in this regional tectonic context, it
would be related to the upper part of the no deformational compressive zone and reverse
earthquakes would appear with the foci located deeper than 2 km. The strain field is
directly related to the permeability tensor due to rock dissolution. Hence, this value
could play an important role for long-term reservoir expected life.

*5.3 Local tectonic field and Induced Seismicity*
The fluid injection into a deep saline aquifer, which is used as CCS, generally increases
the pore pressure. The increasing of the pore pressure migrates from the point of
injection to the whole reservoir. Moreover, changes into the stress field for faults that
are located below the reservoir, could also trigger induced earthquakes (Verdon et al.,
2014). Nevertheless, to understand this possibility and the study the volumetric strain
field spatial distribution is required (Lisle et al., 2009).





The injection of 10 k tons of $CO_2$ in Hontomín (Gastine et al., 2017) represents an
approximate injected volume of $CO_2$ of 5.5 x$10^6$ m$^3$. The earthquake magnitude to this
fluid-injected volume according to the McGarr (2014) and Verdon et al. (2014) could be
M > 5 if there are faults with a minimum size of 10 km and oriented according to the
present stress field within the influence area (N-S extensional faults and NE-SW/NW-
SE strike slip faults). In the case of HPP, there are faults below the reservoir with this
potential earthquake triggering (Alcalde et al., 2014). On the other hand, overpressure
increasing the permeability of the carbonate reservoir along with the pore pressure
variations of about 0.5 MPa, could trigger earthquakes, as well. Stress-drop related to
fluid injections are also reported (Huang et al., 2016).
Le Gallo and de Dios (2018) described two main fault sets affecting the reservoir with
N-S and E-W trend, respectively. According to the present-day stress tensor described
by Herraiz et al. (2000) and Tavani et al. (2011), E-W fault-sets are accommodating
horizontal shortening, which means that the permeability could be low, and N-S faults
could act as strike-slip with trans-tensional component and, hence, higher permeability.
On the other hand, increasing the pore-pressure of E-W faults could reduce de seismic
cycle in these faults. Therefore, special attention has to be paid in microseismicity
related to E-W faults. In this sense, the study of focal mechanisms solutions could
improve the safety management, even for microearthquakes of magnitude lesser than M

692     3.

Moreover, the $CO_2$ lateral diffusion and pressure variation change during the fluid
injection phase, and then the system would relax before to be increased during the next
injection phase. In this context, the intermittent and episodic injection of $CO_2$ could also
trigger earthquakes by the stress-field and fluid pressure variations in short time periods.



## 6. CONCLUSIONS


The application of the analysis for brittle deformation determines the tectonic evolution
of the strain field, applied in Carbon Capture and Sequestration (CCS). The possibility
that pore pressure variations due to fluid injection could change the stress/strain
conditions in the reservoir's caprock, makes the study of the present-day tectonic field
as mandatory for the storage safety operations. In this sense, we have to bear in mind
that this kind of subsurface storage is designed for long-life expectancy, about
thousands of years, and therefore, relevant earthquakes could occur affecting the sealing
and the seepage of $CO_2$, compromising the integrity of the reservoir. Hence, we can
conclude from our analysis the following items:
(1) The study of this tectonic field allows classifying the geometry of the faults to
prevent prone earthquake-related structures and design monitoring seismic network.
(2) The influence area around the facilities of the CCS for studying the active
stress/strain field could reach 20 km from the facility and the tectonic evolution of the
geological history of the reservoir have to be established, adding missing information
from map scale and boreholes. This information could be used from the 3D local
fracture pattern estimation to avoid overpressure for increasing the permeability paths.
Analysis of the stress-drop due to the fluid injection could be combined with this
information to understand potential microseismicity associated with the injection
operations.
(3) In the case of Hontomín Pilot-Plant, we have obtained two strain active tectonic
fields featured as shear deformation. These fields are defined by (a) a local tectonic
strain field with $e_y$, $S_{Hmax}$ striking N50ºE and (b) the regional one defined by $e_y$, $S_{Hmax}$
with N150ºE trend. In this context, strike-slip faults with NE-SW trend, reverse faults
with NW-SE trend and reverse oblique faults oriented E-W, are accumulating present-



day tectonic deformation. Therefore, we propose the monitoring of E-W faults and the
intersection with strike-slip faults, either due to the possibility to make high-permeable
paths for $CO_2$ mobility, or due to the possibility to act as compressional faulting due to
the increasing of the pore pressure during injection.
(4) Both WNW-ESE fault plus N-S and NE-SW directions are the preferential fault
directions for potential fluid leakage. E-W could act as compressive faults.
(5) The Ubierna Fault System represents a tectonically active fault array that could
trigger natural earthquakes as large as M 6 (±0.1), from the empirical relationship of the
total rupture segment (ranging between 12 and 14 km, and the total fault-area rupture,
oscillating between 100 and 150 $km^2$). Despite the lack of instrumental seismicity into
the influence area, we cannot obviate the potential earthquake occurrence within
intraplate areas due to the long- timescale expected-life of the CCS. The heat-flow
values and thermal crust conditions could determine the presence of intraplate
earthquakes with magnitude M > 5, for a long timescale (thousands of years).
The tectonic evolution and kinematics of the west part of the Burgalesa Platform
domain from upper Triassic to present day show a Cretaceous tectonic inversion, local
reverse strain field during the early Oligocene and early Miocene, with a Normal strain
field during the middle Miocene. The active strain field is now defined by a shear
tectonic defined by $e_y$ with N-S trend, activating E-W thrust and right-lateral faults with
WNW- and NW- trend.
Finally, we state that the determination of the active tectonic strain field, the recognition
and study of active faults within the area of influence (20 km), the estimation of the
maximum potential triggered natural earthquake, the modeling of the stress-change
during the fluid injection and stress-drop, probably improve the operations for a secure
storage. In a short future, earthquake scenarios will be the next step: modeling the



Coulomb static stress-changes due to fluid injection and the modeling of intensity maps
of horizontal seismic acceleration.



ACKNOWLEDGEMENTS
This work has been partially supported by the European Project ENOS: ENabling
Onshore $CO_2$ Storage in Europe, H2020 Project ID: 653718 and the Spanish project
3GEO, CGL2017-83931-C3-2-P, MICIU-FEDER. The authors would also thank the
crew of CIUDEN at Hontomín facilities for their kind assistance during our fieldwork.



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
