# Peer review of "Active tectonic field for CO2 Storage management: Hontomín onshore study-case (SPAIN)"

_Solid Earth, 2019_

## Referee Comment (RC1) · Graham Yielding (Referee) · 27 Jan 2020

This manuscript presents a structural analysis of fault data collected at outcrop in the vicinity (<20km) of a potential CO2 storage site. Such data collection is extremely valuable in the context of storage site monitoring, for 2 reasons: 1. Faults within the reservoir and within its overburden may be reactivated by the present-day stress field, allowing fluids to migrate up the fault surfaces and out of the site. A leak of CO2 to the surface could be a serious safety concern. 2. Faults at a greater distance, not directly inside the reservoir location, may be reactivated in the present-day stress field and produce sizeable earthquakes, which could damage the site facilities and also change the stress balance on small intra-reservoir faults, leading to leakage as noted at point 1.

[Figure]

So, data collection of the type undertaken here is very commendable.

However, the bulk of the paper is taken up with an irrelevant (and probably incorrect) attempt to reconstruct the tectonic strain (in tensor form) for numerous periods in the geological past. Such analysis is irrelevant because it has no bearing on the present-day site stability. The strain field (or the causative stress) in the Early Cretaceous has no influence on the susceptibility of faults to slip at the present day. It has been well known for many decades that the orientation of fault planes within the IN SITU stress field is the critical relationship that controls fault reactivation and associated fault permeability (Bott, 1959; Barton et al, Geology, 1995; Morris et al, Geology, 1996). Whilst the fault orientation data is clearly of use here, the authors should concentrate on the present-day (active) stress state, not the "strain field" in the Cretaceous. The present-day stress state is not well constrained in this area: Herraiz et al 2000 suggest SHmax close to N-S, and a quick look at the World Stress Map (Heidbach et al 2016) shows SHmax roughly NE-SW but based on only 3 readings.

For some reason I do not understand, outcrop data collected above the location of the site (HTM17), which comprises almost a quarter of all the data, is "removed". Surely it is the station which is most relevant to the site conditions?

The construction of "strain fields" through geological time is probably incorrect for several reasons. Firstly, by definition, faults which affect a particular stratigraphic unit must be younger than that unit, unless the fault is demonstrably a sedimentary growth fault. Thus faults recorded in Early Cretaceous outcrop might have any age between Early Cretaceous and the present day. The authors appear to partially recognise this because they attribute some strain fields to be superposed "modern paleostrain field" (Fig.9) - but the way this is done seems completely arbitrary. Secondly, the analysis assumes that a fault orientation recorded today is the same as when the fault slipped in the past - but it is well known that rotation about vertical axes is significant in fault systems with a strike-slip component (e.g. Lamb, JSG, 1988; Ron et al, Geology, 1986; and many others). So the inversion results obtained from present-day orientations may

themselves be rotated from their original direction.

For the purpose of contributing to CO2 storage management (as in the manuscript title), I suggest the paper be radically re-written: - The 447 fault data from 32 field stations is excellent - but please include detailed discussion of the data at the site itself (HTM17). - Drop the palaeostrain analysis completely. - Concentrate on the present-day stress field data, and assess the amount of uncertainty in its constraint. - Resolve the present-day stress tensor onto the fault plane orientation data, to illustrate which fault trends have higher slip tendency and hence higher reactivation/leak potential.

Graham Yielding.

―――――――――――――――――

---

## Referee Comment (RC2) · Anonymous Referee #2 · 2 Feb 2020

In their manuscript "Active tectonic field for CO2 storage management: Hontomín onshore study-case (Spain), Pérez-López et al employ fault orientations sampled in the surroundings of the Hontomín site to analyse the past and present stress field. Their results provide an indication of the risk of leakage of the injected CO2 in the reservoir due to tectonic activity or fault reactivation during the storage operations. This seems a reasonable approach, although some aspects of this appraisal are hard to assess with the data presented in the manuscript. I have a number of issues that need to be significantly improved before this paper can be considered up for publication. I outline the major issues here, and attach a detailed list of minor comments that should be addressed by the authors.

General comments 1) The authors should revise several aspects of the manuscript

related to the geological storage of CO2 (GSC), which constitute the ultimate goal of this study.

a. First, it is claimed that this type of studies are not conventionally implemented in monitoring strategies. I would argue the opposite, as the tectonic activity is one of the main factors that need to be considered in the site screening and selection processes according to well-cited best practice guides (e.g. IPCC, 2005; Chadwick et al., 2008; IEAGHG 2009). Fault reactivation and induced seismicity are also major concerns in storage operations, which is reflected in the vast body of literature available (e.g. Nicol et al., 2011; Zoback & Goerlick, 2012; Juanes et al., 2012; Vilarrasa and Carrera 2015; White and Foxall, 2016). The authors fail to put their work in perspective, and I actually suggest them to switch their rationale: these issues are so important that their study is imperative.

b. The terms CCS and CO2 storage are used indistinctively throughout the manuscript (e.g, lines 57, 84, 667), but they are different processes (the former encompassing the latter), and thus should not be confused.

c. I have major concerns about the risk assessment that is proposed in this study. First, the scale of the Hontomín facility is in my opinion overlooked. Hontomín is a pilot storage plant, and as such it was never conceived for the storage of large volumes of CO2. It is extremely unlikely that the injection of 10k tonnes of CO2 would produce any effects in the faults bounding the reservoir, not to mention the reactivation of the regional-scale Ubierna Fault. Also, the volume calculation presented in lines 673-674 is wrong, as it assumes room conditions for the CO2 (i.e. 556.2m^3/ton) instead of reservoir conditions. Besides, the authors cite McGarr's approach to calculate the Mmax of potential fault ruptures, but do not show any of the calculations or the assumptions that they make to claim that M >5 earthquakes are possible as a result of the injection (line 676). This is a very audacious statement (with which I strongly disagree), and as such it must be well documented and presented in the text. In summary, the authors fail to prove how the injection in a small reservoir (which is likely to be compartimentalised from the UFF splay anyways) could produce a distortion so great to re-active the entire 6km long segment.

2) I think that the authors should include some sort of analysis of the statistical significance of fault measurements in their calculations. For example, some ages count with several outcrops (as well as greater areal distribution) and samples than others; how does this affect your paleostress calculations?

3) In terms of the quality of the writing, several portions of the test require a profound re-writing to improve readability. There are remarkable differences in the quality and clarity on some of the sections (for example, sections 3.3 and 3.4 read very well compared to sections 3 and 3.1). I have made suggestions or marked the sentences that particularly require a thorough rephrasing in the attached file, but I encourage the authors to review and polish the entire manuscript or look for potential professional aid in this sense.

References: Chadwick, A., Arts, R., Bernstone, C., May, F., Thibeau, S., and Zweigel, P. 2008. Best practice for the storage of CO2 in saline aquifers. Keyworth, Nottingham: British Geological Survey Occasional Publication No. 14. ISBN: 978-0-85272-610-5 IEA Greenhouse Gas R&D Programme (IEA GHG), "CCS Site Characterisation Criteria", 2009/10, July 2009. IPCC — Intergovernmental Panel on Climate Change, 2005. IPCC Special Report on Carbon Dioxide Capture and Storage. Cambridge University Press, Cambridge, UK – Table 5.3 Juanes, R., Hager, B. H., & Herzog, H. J. (2012). No geologic evidence that seismicity causes fault leakage that would render large-scale carbon capture and storage unsuccessful. Proceedings of the National Academy of Sciences, 109(52), E3623-E3623. Nicol, A., Carne, R., Gerstenberger, M., & Christophersen, A. (2011). Induced seismicity and its implications for CO2 storage risk. Energy Procedia, 4, 3699-3706. Vilarrasa V, Carrera J (2015) Geologic carbon storage is unlikely to trigger large earthquakes and reactivate faults through which CO2 could leak. Proc Natl Acad Sci USA 112(19):5938–5943. White, J. A., & Foxall, W. (2016). Assessing induced seismicity risk at CO2 storage projects: Recent progress and remaining challenges. International Journal of Greenhouse Gas Control,

49, 413-424. Zoback, M. D., & Gorelick, S. M. (2012). Earthquake triggering and large-scale geologic storage of carbon dioxide. Proceedings of the National Academy of Sciences, 109(26), 10164-10168.

Please also note the supplement to this comment:
https://www.solid-earth-discuss.net/se-2019-196/se-2019-196-RC2-supplement.pdf

———————————————————

[Figure]

**Supplement:**

[revised manuscript text omitted]

---

## Author Comment (AC1) · 7 Feb 2020

Letter to Dr. Graham Yielding, related to the interactive comment on "Active tectonic field for CO2 storage management: Hontomín onshore study-case (SPAIN)" by Pérez-López et al. The authors (4 February 2020)

Dear Dr. Yielding, Thank you very much for your time and effort in reviewing the manuscript under the open discussion of Solid Earth and your kind comments. You pointed accurately the main objective of the paper, "the reactivation of faults due to the injection of fluids and the potential for triggering earthquakes, within the vicinity of the CCS (Carbon Capture and Storage), and pilot plant facilities". According to the suggestion of anonymous reviewer #2, hereafter we refer to Geological Storage of CO2

(GSC) instead of CCS. We agree with you that the main goal of the paper was not the paleostrain evolution and global tectonic events recognized in the Basque-Cantabrian area and Duero and Ebro river basins (North part of Spain), but the role of the fault sets affecting the GSC by the present-day strain field. However, the high-quality outcrops in the near-field (<20 km) gave us a good chance to estimate different paleostrain local tensors affecting geological formations at different ages (the Ubierna Fault System and the south-eastward fault-end geometry with the Hontomín Fault). What we mean is we were able to calculate paleostrain tensors affecting Triassic, Cretaceous and Tertiary deposits and we were able to discriminate which of them worked in a particular time interval. Furthermore, the controversy related to the assignment of geological ages to the different strain tensor calculated in geological outcrop is still an open debate. We know well the problems to reconstruct paleostrain fields and how to match these results with large-scale tectonic events throughout the geological evolution of the basin. Of course, stress/strain axis rotations due to different paleogeographic constrain, magnetic field changes, among others, obviously difficult that reconstruction. As a matter of fact, the paleostrain reconstruction is always controversial, more if you take into account that all of our studies are on the local scale. This kind of analysis is always constrained by the quality of the outcrop and the ability to assign strain tensors to large-scale tectonic events affecting the studied area. We have indeed tried to match the paleostrain tensors calculated from slip fault data with those global tectonic events defined by other authors in the area. Perhaps, we have failed to suggest that this is only a local analysis. Consequently, we can accept that the paleostrain evolution could be removed from the final manuscript. Concerning the inclusion of the fault-slip data from the outcrop HTM17, as suggested by Dr. Yielding, we have included a new section. At the beginning of this work, it was a long discussion among the authors about the convenience either to describe this outcrop, having in mind the relevance of the site-effects, or to remove this outcrop from the huge amount of information we had to deal with. Despite that and rethinking from the Dr. Yielding' comment, we will include in the reviewed text the next figures and texts:

Cretaceous Outcrop HTM17 on the Hontomín Pilot Plant This outcrop is located on top of the geological reservoir, in a quarry of Upper Cretaceous limestones. The main advantage of this outcrop is the well-development of striation and carbonate microfibers which yields high-quality data. 105 fault-slip data were measured, with the main orientation striking N75°E; N-50°E; and a conjugate set with N120°E (±10°) trend (Fig. XX). The result of the strain inversion technique shows an extensional field featured by an ey trajectory striking N107°E (±24°) related to an extensional strain field (see the k' diagram in figure XX). Fault sets with ENE-WSW and E-W trend (Fig. XXX) could be reactivated as compressive faults (with lateral component) under the present-day stress field, and NE-SW faults as oblique ones. Minor fault sets with NNE-SSW and NNW-SSE striking could react as extensional faults.

FIG XX

FIG XXX

Besides, we are calculating how the present-day stress tensor affects to each strain tensor solution and fault sets as Yielding' suggestion. Sorry, we have not included it, it takes a while! Finally, we don't agree with Dr. Yielding with the idea concerning the document has to be drastically rewritten. Despite that, thank you very much for your kind comments, revisions, and suggestions, that were properly focused on the aim of the manuscript, the role of the present-day strain field for GSC operations, and which definitively will improve the final manuscript. REFERENCES Herraiz, M., et al. 2000. The recent (upper Miocene to Quaternary) and present tectonic stress distributions in the Iberian Peninsula, Tectonics, 19, 762–786, https://doi.org/10.1029/2000TC900006, 2000. Stich, D., et al. 2006. Kinematics of the Iberia-Maghreb plate contact from seismic moment tensors and GPS observations, Tectonophysics, 426, 295-317. https://doi.org/10.1016/j.tecto.2006.08.004, 2006.

stereogram

**HTM17 N= 105**
strike rose diagram

DIP

RAKE

strike polar plot

polar rake

dip sense rose diagram

**N**

**N**

Right Dihedral

EXTENSION     COMPRESSION
100%  60%  20% 0% 20%  60% 100%

ey solutions

Deym 107º ± 24º

k´ diagram

$10 < K´ < \infty$
$1 < K´ < 10$
$0 < K´ < 1$
$-0,5 < K´ < 0$

$\infty > K´ > -11$
$-11 > K´ > -2$
$-2 > K´ > -1$
$-1 > K´ > -0,5$

**Fig. 1.** Fault data from the outcrop HTM17 located on top of the HPP. See figure 5 for the geographical location. Stereogram plots is lower hemisphere and equal-area net.

**HTM17**

**Reverse fault data**   N = 15

N

a)

N

b)

ey

present-day ey

**Normal fault data**   N = 90

N

a)

N

b)

ey

present-day ey

**Fig. 2.** Figure XXX. Normal and reverse faults stereograms (lower hemisphere and equal area net), and rose diagrams measured in HTM17. Green arrows indicate the orientation of the local paleostrain field. Grey

---

## Author Comment (AC2) · 9 Feb 2020

Letter to the Anonymous referee #2 related to the interactive comment on "Active tectonic field for CO2 storage management: Hontomín onshore study-case (SPAIN)" by Pérez-López et al.

The authors (3 February 2020)

First of all, thank you very much for your time and your effort, and for the constructive review of our work. We really feel that an open discussion improves the scientific results and we are willing to deal with it. Having said that, we answer all the points kindly provided by the Anonymous Referee #2 (hereafter AR2), in the same order he did.

[Figure]

We didn't analyze the past and present "stress" fields, we have analyzed the past and present "strain" fields. Although the orientations of both fields are similar, it is important to highlight the difference in terminology between stress and strain and that we have calculated strains. The Right Dihedral based on fault slip data yields "strain" and "paleostrain" fields. It is assumed that in the general case for the Anderson fault model, ey (the trajectory of maximum strain) is parallel to SHmax (maximum horizontal shortening) being different terms although (e.g. Giner-Robles et al. 2009).

Moreover, our results do not evaluate "the risk" for leakage of $CO_2$ injection neither the possibility of triggering an M5 earthquake as AR2 suggests. RISK is a term that includes HAZARD plus VULNERABILITY plus EXPOSITION, and in our manuscript, we didn't calculate that. In our conclusions, we expose the fault sets and orientation according to the present-day stress orientation which is affecting the caprock and the vicinity. We also describe a low-cost methodology to obtain paleostrain data, fault sets and their relationship with the present-day stress field, and we suggest including this kind of low-cost analysis for long-term geological $CO_2$ storage (GSC).

On the other hand, for estimating the potential seismic triggering of active faults, more analyses have to be carried out, for example, the study of active faulting by tectonic geomorphology and paleoseismology (see McCalpin 1996 and Papanikolaou et al. 2016, for instance). Therefore, the risk assessment for leakage of $CO_2$ is out of the scope of our manuscript. Comment 1a. AR2 said that the analysis we made is common in monitoring strategies for GSC management. We don't know any work about "strain inversion techniques" applied in GSC management so far. All the references suggested by AR2 are devoted to induced seismicity and the tectonic role in GSC management. We do not deny that and we do not claim that we are pioneers in induced seismicity and tectonic studies on GSC. We simply present how the Structural Analysis of fault/slip data can improve the knowledge of the tectonic large-scale fault network for the potential seismic reactivation during fluid injection and time-depend scale for fluid stays. Perhaps we failed to explain more clear our results. We will revise the manuscript to check

this point.

Comment 1b. In this point, we agree with AR2 regarding the terminology about Carbon Capture and Storage (CCS), geological storage even Carbon sequestration (discarded terminology that we have used in the conclusion section). Hence, we will use only one. We agree with the term of geological $CO_2$ storage (GSC) and therefore, we have changed CCS (carbon capture and storage, and other some former term which was included in the manuscript and removed for the revised version) for the term GSC.

Comment 1c. Once again, the "risk assessment" was not performed in our work. This confusion is relevant to solve, it could lead to major mistakes for managing GSC. We insist that we have not carried out any kind of "risk analysis".

On the other hand, the expression "extremely unlike" used by AR2 to describe the potential of triggering M5 earthquakes by active faults near the GSC is not relevant to determine the real seismic hazard. There is no formal reference to this sentence and therefore we have ignored it. We have applied a physic model by the estimation of the total volume injected (yes, we did it in room conditions), from the official data (referenced in the manuscript), and then we have applied the McGarr's (2014) approximation. Taking into account the uncertainties from this analysis, our analysis "should not be regarded as an absolute physical limit", paraphrasing McGarr (2014)' words (page 1, ending sentence of the abstract section). According to McGarr (2014), the utility of the analysis that we have performed is "to predict in advance of a planned injection whether there will be induced seismicity", and in the case of the Hontomín Pilot Plant, by the estimation of the "total injected volume" in a small-scale injection plant (yes, the utility of a small injection plant, as pointed out by Cook et al., 2014). McGarr (2014) applied his approach for three cases: (1) wastewater injection, (2) hydraulic fracturing, and (3) geothermal injection. We have gone one step beyond by including geological storage of $CO_2$. We assume that the pore pressure increases from $CO_2$ injection, in a similar way that wastewater does (originally defined by Frohlich, 2012).

Regarding Lines 673 and 674: We agree with AR2 that we have calculated the volume in "room conditions" for estimating the total injected volume, but we disagree with AR2 that it is a wrong estimation. We have read the work from McGarr (2014), and we did not found a reference that the injected value for the McGarr equation (eq 13) should be in "reservoir conditions". That means that our volume estimation was correct, and simply we have included "room conditions".

Concerning the calculation of the maximum seismic moment by the total injected fluid we did it as follows: We have used the expression Mo(max) (Nm) = G ÂůV (McGarr 2014, eq. 13) Where G for the upper limit is 3 exp10 Pa and V is 5.56 exp3 m3 for the total injected volume (room conditions). The result is 1.65 exp14 Nm (Joules), which corresponds to Mmax = 6.025 of the maximum Richter magnitude by applying the equation Log E = 11.8 + 1.5ÂůM; where Log is the logarithm to the base 10, E is the seismic released energy in Joules, and M the Richter magnitude. We agree with AR2 that we have to include this calculation in the revised manuscript.

Respect to the Ubierna Fault System (UFS), we simply claim that it is necessary to know the seismic cycle of their active segments to know the possibility that small induced seismicity could trigger a natural earthquake by changing the strain conditions. Today, that question is not well-known but as a matter of fact, the study of the strain fields by inversion techniques and the relationships with the active faults in the vicinity is a good approach to get a realistic answer. Up to now, there is not a known model or work analyzing the seismic cycle of the UFS. Our statement about the paleostrain and present-day strain analysis is a suggestion for best practice in the long term geological storage, not a fact. Regarding the "seismic cycle" and missing references, we have included Scholz (2019), one of the most general explanations of a wide-used description of active faults and related references.

Comment 2. AR2 asked a statistical analysis to the fault dataset but the Fault Population analysis in Structural Geology is a statistical analysis itself, so we did it. Perhaps AR2 means a calculation of uncertainties from age outcrops associated with quality

parameters measured on the field. In this sense, we do not find any quality differences between the faults striations measured on fault planes. Any field data with doubt or potential misinterpretation from the field data was directly obviated to avoid biased results. Only high-quality striations and fibers were measured (see annexed figure 1). Moreover, geological ages mapped in the outcrops are enough constrained to be homogenous for the paleostrain reconstructions.

Comment 3. Thank you very much for your time and your effort to improve our English style, although we are not pretty sure that all of the suggestions are correct. Therefore, your suggestions will be revised and included if so. As a non-native English speaker, we always use professional aid. Bearing in mind that the manuscript was written by three different authors and finally homogenized by the first one, the English style could oscillate in different parts of the text. Thanks indeed.

Minor comments in the supplementary doc: (Major questions):

Q1 (abstract). AR2 asks about the "tectonic parameters": They are those parameters which characterize the tectonic field and framework: stress/strain parameters like SHmax, ey, k', R, natural heat-flow, Moho depth, crustal thickness, etc. In this case, the stress/strain ellipsoid and strain trajectories, master faults, tectonic slip rates, were shown in our work.

Q2 (abstract). Did In Salah (Algeria) case analyze the "tectonic strain field"? Please give us a reference.

Q3 (abstract). The reviewer confuses stress with strain again. They could be related in space, but are different concepts, as we said before.

Q4 (page3). Ancient literature does not mean outdated literature. Please, indicate more appropriate literature if so related to the first stages of GSC and why Pearce 2006 is obsolete.

Q5 (Line74). What do you mean by long-term monitoring? We are meaning about the

expected life of the reservoir in geological terms.

Q6 (Line92). Geomechanical models related to the active tectonic field is not the aim of the paper and is out of the scope.

Q7. The election of 20 km for the strain analysis from geological outcrops is well explained. 3. Method and Rationale, section 3.5.

Q8 (Line110). Ok, removed.

Q9 (Line113). Why should we indicate that Hontomín is a pilot plant before the geological framework? We don't know any reason to prioritize this. Geology is the key for underground storage.

Q10 (Line211). Geomorphic markers (misfit).

Q11 (Fig. 5). Some outcrops are close to 20 km but not exactly. Well, despite we use a GPS in the field, some stations were slightly out of 20 km but quite close to have influence in the geological map. We checked if outcrops were located about 20 km but not exactly. This is another reason to include the outcrops in a geological map and showing that they are related to the structures inside the 20 km circle.

Q12 (Line449). We strongly disagree. The tectonic field is relevant and their expressions are the master faults, they accommodate all of the deformations that it generates. The tectonic field and master faults are not independent concepts.

Line592. Reference misprint (Alcalde et al. 2014.). Line631-640. One of the tectonic parameter to be considered is the crustal heat flow (see Q1). The relevance of this paragraph is to highlight that within intraplate areas large earthquakes could appear. Line643. Both strain fields. Line691. The use of focal mechanism solutions is crucial for understanding the failure mechanism in seismic prone areas. This information is world-wide used for the study of seismogenic faulting, modeling, earthquake hazard, seismic wave propagation, design of the seismic network, etc. Line693-696. Well, one thing is the permeability and lateral diffusion due to a single injection, and another thing

is the mechanical behavior of the caprock under episodic injections.

REFERENCES Cook, Peter, Rick Causebrook, John Gale, Karsten Michel, Max Watson. (2014). What have we learned from small-scale injection projects? Energy Procedia 63, 6129 – 6140. Frohlich, C. (2012). Two-year survey comparing earthquake activity and injection-well locations in the Barnett Shale, Texas. PNAS, vol. 109 (35), 13934 – 13938. Giner-Robles J.L., Pérez-López, R., Rodríguez-Pascua, M.A., Martínez-Díaz, J.J. and González-Casado, J.M. (2009). Present-day stress field on the South American slab underneath the Sandwich Plate (Southern Atlantic Ocean). In: James, K. H., Lorente, M. A. & Pindell, J. L. (eds), The Origin and Evolution of the Caribbean Plate. Geological Society of London, Special Publications 328, 153- 165. DOI: 10.1144/SP328.6 McCalpin J.P. (1996). Chapter 9 Application of paleoseismic data to seismic hazard assessment and neotectonic research. International Geophysics 62: 439-493. McGarr, A. (2014), Maximum magnitude earthquakes induced by fluid injection. J. Geophys. Res. Solid Earth,119, 1008–1019, DOI:10.1002/2013JB010597. Papanikolaou, Ioannis D., Ronald van Balen, Pablo G. Silva, Klaus Reicherter. (2016). Geomorphology of Active Faulting and seismic hazard assessment: New tools and future challenges. Geomorphology, DOI: 10.1016/j.geomorph.2015.02.024. Pearce, J. M. (2005). What can we learn from Natural Analogues? An overview of how analogues can benefit the geological storage of CO2, in: Advances in the Geological Storage of Carbon Dioxide, edited by Lombardi, S., Altunina, L. K., and Beaubien, S. E., Springer, Dordrecht, The Netherlands, 129–139. Scholz, C. (2019). The seismic cycle. The Mechanics of Earthquakes and Faulting (pp. 228-277). Cambridge: Cambridge University Press.

[Figure]

[Figure]

**Fig. 1.** Example of high-quality fibers on a Cretaceous fault plane measured in the station HTM3 (see manuscript for the geographical location). You can observe as the fibers lineation change with th

---

## Author Comment (AC3) · 9 Feb 2020

Sorry for the first post... the mathematical calculations were automatically changed by the letter font

here you are a clear text:

We have used the expression Mo(max) (Nm) = G x V (McGarr 2014, eq. 13) Where G for the upper limit is 3 x 10exp10 Pa and V is 5.56 x 10exp3 m3 for the total injected volume (room conditions). The result is 1.65 x 10exp14 Nm (Joules), which corresponds to Mmax = 6.025 of the maximum Richter magnitude by applying the equation Log E = 11.8 + 1.5 x M; where Log is the logarithm to the base 10, E is the seismic released energy in Joules, and M the Richter magnitude.

---

## Author Response (AR3)

**Dear Editor,**

We have revised our original manuscript titled: "Active tectonic field for $CO_2$ Storage management: Hontomín onshore study-case (SPAIN)", by Pérez-López et al., according to the revisions suggested by Pr. Graham Yielding and an anonymous reviewer (AR2) under the Open Discussion of Solid Earth.

We thank both reviewers their generous contribution. Firstly, we have revised Mr. Yielding's comments because this revision was more complex. Secondly, the anonymous AR2 comments were also faced. Definitively, both remarks have improved the original manuscript and focused the main goal, the study of the active tectonics applied in $CO_2$ storage, on the text. We hope that this revision response enough both reviewers and the SE editor. Both reviewers are thanked for their opportune and successful contributions.

*The authors, March, 2020.*

**P.S.** We have detected some minor mistakes and typos not noted by the reviewers that we have removed.

Answers step-by-step:

**Dear Dr. Yielding,**

Thank you very much for your time and effort in reviewing the manuscript under the open discussion of Solid Earth and your kind comments. You pointed accurately the main objective of the paper, "the reactivation of faults due to the injection of fluids and the potential for triggering earthquakes, within the vicinity of the CCS (Carbon Capture and Storage), and pilot plant facilities". According to the suggestion of anonymous reviewer #2, hereafter we refer to Geological Storage of $CO_2$ (GSC) instead of CCS.

We agree with you that the main goal of the paper was not the paleostrain evolution and global tectonic events recognized in the Basque-Cantabrian area and Duero and Ebro river basins (North part of Spain), but the role of the fault sets affecting the GSC by the present-day strain field. The high-quality outcrops in the near-field (<20 km) gave us a good chance to estimate different paleostrain local tensors affecting geological formations at different ages.

On the other hand, the controversy related to the assignment of geological ages to the different strain tensors calculated in geological outcrop is still an open debate. We well-know the problems to reconstruct paleostrain fields and how to match these results with large-scale tectonic events throughout the geological evolution of a geological basin. Of course, stress/strain axis rotations due to different paleogeographic constrains, magnetic field changes, among others, obviously difficult that reconstruction. As a matter of fact, the paleostrain reconstruction is always controversial, more if you take into account that all of our study is restricted to the local scale. This kind of analysis is always constrained by the quality of the outcrop and the ability to assign strain tensors to large-scale tectonic events affecting the studied area. We have indeed tried to match the paleostrain tensors calculated from slip fault data with those global tectonic events defined by other authors in the area. Perhaps, we have failed to suggest that this is only a local analysis.

**Consequently, we accept to remove the paleostrain evolution from the manuscript. Instead, we have included the Mohr-Coulomb failure criterion applied to the fault pattern affecting the outcrop located in the Hontomín Pilot Plant (subsection 5.1 within the section 5. Discussion). Therefore, original figure 8 and figure 9 were removed and new figures were included: Figs. 8, 9, 10 and 11.**

Ancient Fig. 8 (**REMOVED**): Upper left frame: Synthesis of the K'- map obtained from Giner-Robles et al. (2018) for the whole Iberian Peninsula from focal mechanism solutions. HPP is located between a triple junction of K' defined by compression towards the north, extension to the southeast and strike-slip to the west. Black dots are the earthquakes with focal mechanism solutions used. Sketches represent the ey, $S_{Hmax}$ trajectories obtained from the outcrops for early Triassic, Early Cretaceous, Late Cretaceous, Early Oligocene, Early – Middle Miocene and Present-day strain field from Herraiz et al. (2000). Main structures activated under the strain field defined are also included.

Ancient Fig. 9 (**REMOVED**): Tectonic field evolution of the Burgalesa Platform domain (20-km radius circle centred at HPP), interpreted from the paleostrain analysis. The regional tectonic field from other authors are also included. Ages were estimated from the HTM outcrops affecting geological well-dated deposits. N: extensional faulting; R: compressive faulting; SS: strike-slip faulting. Red and white means local paleostrain field, and red with white dots means superposed modern paleostrain field.

Concerning the inclusion of the fault-slip data from the outcrop HTM17, as suggested by Dr. Yielding, we have incorporated a new section. At the beginning of this work, it was a long discussion among the authors about the convenience either to describe this outcrop, having in mind the relevance of the site-effects, or to remove this outcrop from the huge amount of information we had to deal with. We have included the section 4.4 Cretaceous Outcrop HTM17 on the Hontomín Pilot Plant and several new figures.

New Figs. 8 and 9: Results of the paleostrain analysis of HTM17

[Figure]

**FIGURE 8**          **FIGURE 9**

New Fig. 8. Fault data from the outcrop HTM17 located on top of the HPP. See figure 5 for the geographical location. Stereogram plot is lower hemisphere and Schmidt net.

New Fig. 9. Normal and reverse faults stereograms (lower hemisphere and Schmidt net), and rose diagrams measured in HTM17. Green arrows indicate the orientation of the local paleostrain field. Grey arrows indicate the orientation of the present-day regional stress field (Herraiz et al., 2000).

Besides, we have calculated how the present-day stress tensor affects to each strain tensor solution and fault sets as Yielding' suggestion. (Figure 10, Mohr-Coulomb failure criterion).

[Figure]

Figure 10. Mohr-Coulomb failure analysis for the fault-slip data measured in HTM17 under the present-day stress tensor determined by Herraiz et al. (2000). Red dots are faults reactivated, and green and orange dots are located within the stable zone. Red rose diagram shows the orientation of reactivated faults, between N-S to N60°E and from N115°E to N180°E. Green rose diagram shows the fault orientation for faults non-reactivated under the active tress field within the area. See text for further details.

Some sentences were modified from the abstract, introduction and other sections of the manuscript, concerning that HTM17 was not included here.

Finally, we don't agree with Dr. Yielding with the idea concerning the document has to be drastically rewritten. It is true that we have to include a new section describing the HTM17 outcrop and two new figures with the main data, and it is true again that we have rearranged the discussion section, by including the Mohr-Coulomb failure analysis applied to the HTM17 fracture pattern. However, the main analysis and focus of the manuscript is the same.

Nevertheless, we would like to thank to Pr. Yielding for putting the focus of our manuscript in the fault reactivation under fluid injection in GCE storage. We were some naïve to include a more complex analysis like the paleostrain reconstruction and matching with major tectonic events.

Thank you very much for your kind comments, revisions, and suggestions, that were properly focused on the aim of the manuscript, the role of the present-day strain field for GSC operations, and which definitively will improve the final manuscript.

[Figure]

**Figure 11**. a) Stereogram and poles of fault sets (HTM17) reactivated under the present-day stress field suggested by Herraiz et al. (2000). b) Right-Dihedral of the reactivated fault sets. c) K'-strain diagram showing the type of fault for each fault-set.

**NEW REFERENCES INCLUDED:**

Allmendinger, R. W., Cardozo, N. C., and Fisher, D.: Structural Geology Algorithms: Vectors & Tensors: Cambridge, England, Cambridge University Press, 289 pp, 2012.

De Vicente, G. Muñoz, A., Giner, J.L.: Use of the Right Dihedral Method: implications from the Slip Model of Fault Population Analysis, Rev. Soc. Geol. España, 5(3-4), 7-19. 1992.

Goodman, R. E.: Introduction to Rock Mechanics, 2nd Edition. John Wiley & Sons, Inc., New York. 576 pp., 1989.

Labuz, J. F., Zang, A.: Mohr–Coulomb Failure Criterion. Rock Mech. Rock Eng., 45, 975–979, https://doi.org/10.1007/s00603-012-0281-7, 2012.

Pan, P., Wu, Z., Feng, X., Yan, F.: Geomechanical modeling of $CO_2$ geological storage: A review. J. Rock Mech. and Geotech. Eng., 8, 936-947, http://dx.doi.org/10.1016/j.jrmge.2016.10.002, 2016.

Xu, S.-S., A.F. Nieto-Samaniego, S.A. Alaniz-Álvarez: 3D Mohr diagram to explain reactivation of pre-existing planes due to changes in applied stresses. Rock Stress and Earthquakes – Xie (ed.), 739-745 p, 2010.

**Answer to Anonymous referee #2 (AR2) related to the interactive comment on "Active tectonic field for CO$_2$ storage management: Hontomín onshore study-case (SPAIN)" by Pérez-López et al.**

*The authors (March, 2020)*

First of all, thank you very much for your time and your effort, and for the constructive review of our work. We really feel that an open discussion improves the scientific results and we are willing to deal with it.

As detailed below, we answer all the points kindly provided by the Anonymous Referee #2 (hereafter AR2), in the same order he did.

We didn't analyze the past and present "stress" fields, we have analyzed the past and present "strain" fields. Although the orientations of both fields are similar, it is important to highlight the difference in terminology and concept between stress and strain. We have calculated strains. The Right Dihedral based on fault slip data yield "strain" and "paleostrain" fields. It is assumed that in the general case for the Anderson fault model, ey (the trajectory of maximum strain) is parallel to S$_{Hmax}$ (maximum horizontal shortening) being different terms although (e.g. Giner-Robles et al. 2009). Therefore, the term strain is well used across the manuscript.

Our results do not evaluate "the risk" for leakage of CO$_2$ injection neither the possibility of triggering a M5 earthquake as AR2 suggests. RISK is a term that includes HAZARD plus VULNERABILITY plus EXPOSITION, and in our manuscript, we didn't calculate that. In our conclusions, we expose the fault sets and orientation according to the present-day stress orientation which are affecting the caprock and the vicinity. We also describe a low-cost methodology to obtain paleostrain data, fault sets and their relationship with the present-day stress field, and we suggest including this kind of low-cost analyses for long-term geological CO$_2$ storage (GSC).

On the other hand (**we have removed this expression from the manuscript obeying the suggestion of AR2**), for estimating the potential seismic triggering of active faults, more analyses have to be carried out, for example, the study of active faulting by tectonic geomorphology and paleoseismology (see McCalpin 1996 and Papanikolaou et al. 2016, for instance). Therefore, the risk assessment for leakage of CO$_2$ is out of the scope of our manuscript.

**Comment 1a.** AR2 said that the analysis we made is common in monitoring strategies for GSC management. We don't know any work about "strain inversion techniques" applied in GSC management so far. All the references suggested by AR2 are devoted to induced seismicity and the tectonic role in GSC management. We do not claim that we are pioneers in induced seismicity and tectonic studies on GSC. We simply present how the Structural Analysis of fault/slip data can improve the knowledge of the tectonic large-scale fault network for the potential seismic reactivation during fluid injection and time-depend scale for fluid stays. We have reinforced this idea by including this sentence at the end of the introduction section.

**Comment 1b**. **In this point, we agree with AR2** regarding the terminology about Carbon Capture and Storage (CCS), geological storage even Carbon sequestration (discarded terminology that we have used in the conclusion section). Hence, we will use only one. We agree with the term of geological $CO_2$ storage (GSC) and therefore, **we have changed CCS (carbon capture and storage, and other some former term which was included in the manuscript and removed for the revised version) for the term GSC**.

**Comment 1c.** Once again, the "*risk assessment*" was not performed in our work. This confusion is relevant to solve, it could lead to major mistakes for managing GSC. We insist that we have not carried out any kind of "risk analysis".

On the other hand, the expression used by AR2: "extremely unlike", to describe the potential of triggering M5 earthquakes by active faults near the GSC is not relevant to determine the real seismic hazard. There is no formal reference to this sentence and therefore we have ignored it.

We have applied a physic model to estimate the total volume injected (yes, we did it in room conditions), from the official data (referenced in the manuscript), and then we have applied the McGarr's (2014) approximation. Taking into account the uncertainties, our analysis "*should not be regarded as an absolute physical limit*", paraphrasing McGarr (2014)' words (page 1, ending sentence of the abstract section). According to McGarr (2014), the utility of the analysis that we have performed is "to predict in advance of a planned injection whether there will be induced seismicity", and in the case of the Hontomín Pilot Plant, by the estimation of the "total injected volume" in a small-scale injection plant (yes, the utility of a small injection plant, as pointed out by Cook et al., 2014).

McGarr (2014) applied his approach for three cases: (1) wastewater injection, (2) hydraulic fracturing, and (3) geothermal injection. We have gone one step beyond by including geological storage of $CO_2$. We assume that the pore pressure increases from $CO_2$ injection, in a similar way that wastewater does (originally defined by Frohlich, 2012). **We have included this paragraph in the revised manuscript**.

Regarding **Lines 673 and 674**: **We agree with AR2** that we have calculated the volume in "room conditions" for estimating the total injected volume, but we disagree with AR2 that it is a wrong estimation. We have read the work from McGarr (2014), and we did not found a reference that the injected value for the McGarr equation (eq 13) should be in "reservoir conditions". That means that our volume estimation was correct, and simply we have included "room conditions".

**Regarding the calculation of the maximum seismic moment Mo(max), originated by the total injected fluid, we included the next paragraph:**

*We have applied a physic model to estimate the total volume injected (room conditions), and then we have applied the McGarr's (2014) approximation. The injection of 10 k tons of $CO_2$ in Hontomín (Gastine et al., 2017), represents an approximated injected volume of $CO_2$ of 5.56 $x10^6$ $m^3$ (room conditions). We have used the expression Mo(max) (Nm) = G $\cdot \Delta V$ (McGarr 2014, eq. 13), where G is the modulus of rigidity and for the upper limit is 3 x $10^{10}$ Pa, and $\Delta V$ is the*

*total injected volume (in room conditions). The result is Mo(max) equal to 1.67 x 10$^{17}$ Nm (Joules), which corresponds to a maximum seismic moment magnitude Mw (max) = 5.45, by applying the equation Mw = (Log Mo(max) − 9.05)/1.5 from* Hanks and Kanamori (1979)*; where Log is the logarithm to the base 10.*

**We agree with AR2 that we have to include this calculation in the revised manuscript**. Also, **it was a misprint of the result, the correct number is Mw= 5.45 instead of M= 6.01, by using the seismic moment magnitude instead of the Richter Magnitude. We have solved this and give thanks to AR2 for reviewing the math operations.**

Regarding the Ubierna Fault System (UFS), we simply claim that it is necessary to know the seismic cycle of their active segments to know the possibility that small induced seismicity could trigger a natural earthquake by changing the strain conditions. Today, that question is not well-known but as a matter of fact, the study of the strain fields by inversion techniques and the relationships with the active faults in the vicinity is a good approach to get a realistic answer. Up to now, there is not a known model or work analyzing the seismic cycle of the UFS. Our statement about the paleostrain and present-day strain analysis is a suggestion for best practice in the long term geological storage, not a fact. Regarding the "seismic cycle" and missing references, we have included Scholz (2018), one of the most general explanations of a wide-used description of active faults and related references.

**Comment 2.** AR2 asked a statistical analysis to the fault dataset but the Fault Population analysis in Structural Geology is a statistical analysis itself, so we did it. Perhaps AR2 means a calculation of uncertainties from age outcrops associated with quality parameters measured on the field. In this sense, we do not find any quality differences between the faults striations measured on fault planes. Any field data with doubt or potential misinterpretation from the field data was directly obviated to avoid biased results. Only high-quality striations and fibers were measured (see annexed figure 1). Moreover, geological ages mapped in the outcrops are enough constrained to be homogenous for the paleostrain reconstructions.

**Comment 3.** Thank you very much for your time and your effort to improve our English style, although we are not pretty sure that all of your suggestions are correct. Anyway, all of your suggestions will be revised and included if so. As a non-native English speaker, we always use professional aid. Bearing in mind that the manuscript was written by three different authors and finally homogenized by the first one, the English style could oscillate in different parts of the text. Thanks indeed.

*Minor comments in the supplementary doc: (Major questions, English grammar questions are no answered here). In total, 17 questions were faced and answered. Line number is for the annotated text of the supplementary doc by the reviewer.*

**Q1 (abstract).** AR2 asks about the "tectonic parameters": They are those parameters which characterize the tectonic field and framework: stress/strain parameters like $S_{Hmax}$, ey, k', R, natural heat-flow, Moho depth, crustal thickness, etc. In this case, the stress/strain ellipsoid and strain trajectories, master faults, tectonic slip rates, were showed in our work. **WE HAVE INCLUDED THIS SENTENCE.**

**Q2 (abstract).** Did In Salah (Algeria) case analyze the "*tectonic strain field*"? Please give us a reference.

**Q3 (abstract).** The reviewer confuses **stress** with **strain** again. They could be related in space, but are different concepts, as we said before. (see comment 1c.)

**Q4 (page3).** Ancient literature does not mean outdated literature. Please, indicate more appropriate literature if so related to the first stages of GSC and why Pearce 2006 is obsolete.

**Q5 (Line74).** What do you mean by long-term monitoring? We are meaning about the expected life of the reservoir in geological terms, from hundreds to thousand years.

**Q6 (Line92).** Geomechanical models related to the active tectonic field is not the aim of the paper and is out of the scope.

**Q7**. The election of 20 km for the strain analysis from geological outcrops is well explained. (3. Method and Rationale, section 3.5).

**Q8 (Line110). Ok, removed**.

**Q9 (Line113).** Why should we indicate that Hontomín is a pilot plant before the geological framework? We don't know any reason to prioritize this. Geology is the key for underground storage.

**Q10** (**Line211**). **Geomorphic markers (misfit)**.

**Q11 (Fig. 5).** Some outcrops are close to 20 km but not exactly. Well, despite we use a GPS in the field, some outcrops were slightly out of 20 km but quite close to have influence in the geological map. We checked if outcrops were located about 20 km but not exactly. This is another reason to include the outcrops in a geological map.

**Q12 (Line449).** We strongly disagree. The tectonic field is relevant and their expressions are the master faults, they accommodate all of the deformations that it generates. The tectonic field and master faults are not independent concepts.

**Line 592. Reference misprint removed** (Alcalde et al., 2014.).

**Line 631-640**. One of the tectonic parameter to be considered is the crustal heat flow (see Q1). The relevance of this paragraph is to highlight that within intraplate areas large earthquakes could appear.

**Line 643**. Both strain fields.

**Line 691.** The use of focal mechanism solutions is crucial for understanding the failure mechanism in seismic prone areas. This information is world-wide used for the study of seismogenic faulting, modeling, earthquake hazard, seismic wave propagation, design of the seismic network, etc.

**Line 693-696.** Well, one thing is the permeability and lateral diffusion due to a single injection, and another thing is the mechanical behavior of the caprock under episodic injections. Both things have to be considered.

Minor remarks like the use of cap letters in Induced Seismicity and minor English grammar suggestions have been included in the revised manuscript.

**NEW REFERENCES**

Frohlich, C.: Two-year survey comparing earthquake activity and injection-well locations in the Barnett Shale, Texas. PNAS, vol. 109 (35), 13934 – 13938. 2012.

Hanks, T.C., Kanamori, H.: A Moment Magnitude Scale, J. Geophys. Res., 84 (B5), 2348–2350, https://doi.org/10.1029/JB084iB05p02348, 1979.

Scholz, C.: The seismic cycle. In The Mechanics of Earthquakes and Faulting (pp. 228-277). Cambridge: Cambridge University Press. 2018.

Active tectonic field for CO$_2$ Storage management: Hontomín onshore study-case (SPAIN)

Raúl Pérez-López[1], José F. Mediato[1], Miguel A. Rodríguez-Pascua[1], Jorge L. Giner-Robles[2],

Adrià Ramos[1], Silvia Martín-Velázquez[3], Roberto Martínez-Orío[1], Paula Fernández-

Canteli[1]

1. IGME – Instituto Geológico y Minero de España – Geological Survey of Spain. C/Ríos Rosas 23,
Madrid 28003 – SPAIN. Email: r.perez@igme.es, jf.mediato@igme.es, ma.rodriguez@igme.es,
a.ramos@igme.es, ro.martinez@igme.es, paula.canteli@igme.es
2. Departamento de Geología y Geoquímica. Facultad de Ciencias. Universidad Autónoma de Madrid.
Campus Cantoblanco, Madrid. SPAIN. Email: jorge.giner@uam.es
3. Universidad Rey Juan Carlos. Email: silvia.martin@urjc.es

*Abstract*

*One of the concerns of underground CO$_2$ onshore storage is the triggering of Induced*

*Seismicity and fault reactivation by pore pressure increasing. Hence, a comprehensive*

*analysis of the tectonic parameters involved in the storage rock formation is mandatory*

*for safety management operations. Unquestionably, active faults and seal faults*

*depicting the storage bulk are relevant parameters to be considered. However, there is*

*a lack of analysis of the active tectonic strain field affecting these faults during the CO$_2$*

*storage monitoring. The advantage of reconstructing the tectonic field is the possibility*

*to determine the strain trajectories and describing the fault patterns affecting the*

*reservoir rock. In this work, we adapt a methodology of systematic geostructural*

*analysis to the underground CO$_2$ storage, based on the calculation of the strain field*

*and defined by the strain field from kinematics indicators on the fault planes ($e_y$ and $e_x$*

*for the maximum and minimum horizontal shortening respectively). This methodology*

*is based on statistical analysis of individual strain tensor solutions obtained from fresh*

*outcrops from Triassic to Miocene. Consequently, we have collected 447 fault data in*

*32 field stations located within a 20 km radius. The understanding of the fault sets role*

*for underground fluid circulation can also be established, helping for further analysis*

*about CO$_2$ leakage and seepage. We have applied this methodology to Hontomín*


[revised manuscript text omitted]
 Mo(max) w (max) = 65.345, of the maximum Richter magnitude by applying the equation Mw =

(Log EMo(max) = 11.8 + 1.5 M 9.05)/1.5 from Hanks and Kanamori (1979); where

Log is the logarithm to the base 10, E is the seismic released energy in Joules, and M

the Richter magnitude. McGarr (2014) applied this approach for three cases: (1)

wastewater injection, (2) hydraulic fracturing, and (3) geothermal injection. We propose to include this approach for fluid injection related to geological storage of $CO_2$. We assume that the pore pressure increases from $CO_2$ injection in a similar way that wastewater does (originally defined by Frohlich, 2012). According to McGarr (2014), the utility of the analysis that we have performed is "to predict in advance of a planned injection whether there will be induced seismicity", and in the case of the Hontomín

[revised manuscript text omitted]

**Dear Dave,**

Thank you very much for your comments. We agree with your recommendations and consequently, we have included the volume estimation of the total injected brine plus $CO_2$ in reservoir conditions. Also, a brief explanation about the orange dots obtained in M-C analysis is included in the caption of Figure 10 and finally, we have discussed the role of the slip and dilatation tendency as suggested.

*Best regards,*

*The authors*

DETAILED COMMENTS:

*I would like to insist on one more change: please calculate the volume of the $CO_2$ at the reservoir conditions (depth/pressure, temperature) and then re-calculate the maximum predicted moment for comparison to the room conditions value. This is important, as it will surely be less.*

**New calculation and text:**

*We have applied a physic model to estimate the total volume injected (room conditions) and in reservoir conditions. Then we have applied the McGarr's (2014) approximation of the maximum expected seismic moment for induced earthquakes. The injection of 10 k tons of $CO_2$ in Hontomín (Gastine et al., 2017), represents an approximated injected volume of $CO_2$ of 5.56 $x10^6$ $m^3$ (room conditions, pressure of 1 bar and temperature of 20 °C). The P/T conditions at the bottom of the wells have a maximum value close to 190 bar (Ortiz et al., 2015; Kovacs et al., 2015), although oscillating between 125 and 170 bar and with a maximum temperature close to 58 °C. Kovacs et al. (2015) pointed out a pressure gradient 0,023 MPa/m and a thermal vertical gradient of 0.033 °C/m, which would correspond to a pressure of 357 bar and 51 °C at 1,550 m depth. P/T bottom values obtained from the observational wells (HA and HI) by Ortiz et al. (2015) and Kovacs et al. (2015), were 170 bar and 42 °C respectively.*

*We have used the general law for gases $P_1*V_1/T_1 = P_2*V_2/T_2$. Therefore, the total injected volume in reservoir conditions according to the parameters observed at the bottom of the wells are, $P_1= 1.01$ bar, $T_1= 20$ °C, $V_1= 5.56 \times 10^6$ $m^3$, $P_2= 170$ bar and $T_2= 42$ °C. Hence, the total volume of injected $CO_2$ plus brine is $6.94 \times 10^4$ $m^3$.*

*McGarr (2014) empirically determined the maximum seismic moment related to a volume increasing by underground injection. The expression is Mo(max) (Nm) = $G \cdot \Delta V$ (McGarr 2014, eq. 13), where G is the modulus of rigidity and for the upper limit is $3 \times 10^{10}$ Pa, and $\Delta V$ is the total injected volume (we have used the total injected volume in reservoir conditions). The result is Mo(max) equal to $2.1 \times 10^{15}$ Nm, which corresponds to a maximum seismic moment magnitude Mw (max) = 4.2, by applying the equation Mw = (Log Mo(max) − 9.05)/1.5 from Hanks and Kanamori (1979); where Log is the logarithm to the base 10.*

*The Mohr-Coulomb diagrams in relation to Reviewer 1 comments are good, but you might want to look at slip and dilation tendency (Morris et al., 1996; Ferrill et al., 1999) too, if not for this paper then for the future.*

We guess that the complete reference is Morris, Alan, David A. Ferrill and D. Brent Henderson: Slip-tendency analysis and fault reactivation. Geology, 24,275-278. doi: [http://doi.org/10.1130/0091-7613(1996)024<0275:STAAFR>2.3.CO;2](http://doi.org/10.1130/0091-7613(1996)024<0275:STAAFR>2.3.CO;2), 1996.

Yes, we are aware of this type of analysis for fault reactivation and it is totally complementary with our work. In fact, applying this analysis is similar because both analyses are based on the same concepts: stress tensor (R and K) and the fault orientation. Moreover, if you see the original paper of Morris et al. (1996), Fig. 4A shows $\sigma_2$ with N-S trend and $\sigma_3$ with E-W trend, under a strike-slip stress tensor. In this context, the most likelihood fault orientation to slip reactivation is N-S with dip greater than 59°. These values are similar to our results in Hontomin, with the same stress tensor and fault orientation than Morris et al. (1996), but in the Basque Cantabria basin instead of Yucca Mountains. The most interesting thing is that results are similar in both cases. Anyway, we can perform this kind of analysis combined with our analysis but as you say, for future works. Even so, we have mentioned this type of analysis in Discussion section of fault reactivation. Thank you for the suggestion.

**Included text:**

*We propose as a complementary and future work, a combined analysis between the fault population analysis and the slip-tendency analysis (Morris et al. 1996), which could improve and discriminate those fault sets most likely to be reactivated under an active stress field. Although both analyses (Fault Population and slip-tendency) are based on the stress tensor and the orientation of fault traces, the slip-tendency also includes rock strength values obtained from the "in situ" tests.*

*Also, what are the yellow data on the M-C plot? They do not have a corresponding rose plot...*

*The yellow data in the M-C diagrams are referred to those planes close to be reactivated, and potentially reactivated by increasing the pore pressure.*

**We have included this sentence in the caption of Figure 9.**

REFERENCES

[revised manuscript text omitted]